# Hysteresis of tropical forests in the 21st century

Arie Staal [1,2✉], Ingo Fetzer [1], Lan Wang-Erlandsson [1], Joyce H. C. Bosmans[3], Stefan C. Dekker [2], Egbert H. van Nes[4], Johan Rockström[1,5] & Obbe A. Tuinenburg [2]

Tropical forests modify the conditions they depend on through feedbacks at different spatial scales. These feedbacks shape the hysteresis (history-dependence) of tropical forests, thus controlling their resilience to deforestation and response to climate change. Here, we determine the emergent hysteresis from local-scale tipping points and regional-scale forest-rainfall feedbacks across the tropics under the recent climate and a severe climate-change scenario. By integrating remote sensing, a global hydrological model, and detailed atmospheric moisture tracking simulations, we find that forest-rainfall feedback expands the geographic range of possible forest distributions, especially in the Amazon. The Amazon forest could partially recover from complete deforestation, but may lose that resilience later this century. The Congo forest currently lacks resilience, but is predicted to gain it under climate change, whereas forests in Australasia are resilient under both current and future climates. Our results show how tropical forests shape their own distributions and create the climatic conditions that enable them.

[1] Stockholm Resilience Centre, Stockholm University, Stockholm, Sweden. [2] Department of Environmental Sciences, Copernicus Institute of Sustainable Development, Utrecht University, Utrecht, The Netherlands. [3] Department of Environmental Sciences, Radboud University, Nijmegen, The Netherlands. [4] Aquatic Ecology and Water Quality Management Group, Wageningen University, Wageningen, The Netherlands. [5] Potsdam Institute for Climate Impact Research, Potsdam, Germany. ✉email: ariestaal@gmail.com

Tropical forests are important regulators of the global climate[1] and the effects of their loss could cascade through the Earth system[2]. Furthermore, they mediate their regional climate by enhancing atmospheric moisture recycling and thereby enhancing rainfall levels at seasonal to annual time scales[3]. These functions depend on several feedback mechanisms which, at the same time, affect and are affected by the distributions of tropical forests[4]. These feedbacks operate at different spatial scales. At a local scale (~1 km), the distribution of continuous values of tree cover ('forest cover' from here on) is distinctly bimodal[5,6]. In other words, generally, either a fully covered forest or a sparsely covered nonforest (savanna or grassland) is found. This pattern, which is consistent across the tropics and under a range of climates, cannot be explained by bimodality in environmental variables[5]. Instead, forest cover bimodality is understood as a result of locally acting feedback processes that can generate alternative stable states[6]. In case of such bistability, disturbances can make the system tip[7], with fire as the most likely mechanism that can make a tropical forests tip to a state of low cover[5,8,9] (Fig. 1a). Crucially, part of the distributions of tropical forests on the planet cannot simply be determined based on the present climate. Instead, among the many factors that affect present forest extent is past forest extent; in other words, the system exhibits hysteresis. Moreover, the importance of past forest extent could be amplified by forest–rainfall interactions.

At a regional scale (100–1000 km[10,11]), tropical forests enhance rainfall[3]. When trees photosynthesize, they extract soil moisture or groundwater and release it to the atmosphere. In this way, up to a certain point, trees can maintain photosynthesis during droughts[12], while they alleviate these droughts themselves[11]. This increase in evapotranspiration can enhance rainfall over large areas[11,13], especially since water can re-evaporate and rain down multiple times[14]. This forest–rainfall feedback is a self-stabilising mechanism that elevates regional rainfall levels and reinforces the hysteresis of tropical forests[15] (Fig. 1b).

Improved data availability from remote sensing and advances in high-detailed hydrological and atmospheric simulations has enabled significant steps in our understanding of these feedbacks[5,6,13,15]. However, the tropical forest hysteresis that emerges from the combination of local-scale tipping points and regional feedback under current and future climates remains unknown. Here, we report the range of possible stable configurations of tropical forest under recent (2003–2014) and projected end-of-century (2071–2100) climatic conditions, based on (1) remote-sensing-based estimates of local hysteresis as delimited by local-scale tipping points, (2) high-resolution hydrological and atmospheric moisture tracking simulations, and (3) rainfall projections from a severe climate-change scenario (SSP5-8.5) in Coupled Model Intercomparison Project phase 6 (CMIP6) model runs. We thus map the range of possible forest distributions now and under severe climate change. We find that the regional-scale forest–rainfall feedback expands this range across the tropics, but especially in the Amazon. Projected rainfall reductions may decrease the minimal extent of the Amazon forest, while projected rainfall increases may expand the minimal extent of the Congo forest. In Australasia, the forest–rainfall feedback has relatively small effects on forest distributions under both current and projected climates.

## Results

**Stability of tropical forests**. First, we estimate the patterns of stability of tropical forests in the latitudinal band 15°N–35°S across all continents based on the recent climate (refs. [5,6]; see 'Methods'). Forest cover distributions (excluding human-used areas, water bodies, and bare ground; see 'Methods') indicate that forest cover in South America is bistable between mean annual rainfall levels of 1250–2050 mm per year; within this range, both forests and a savanna-like nonforested state are found. For Africa we find this bistability between 1350–2050 mm per year, and in

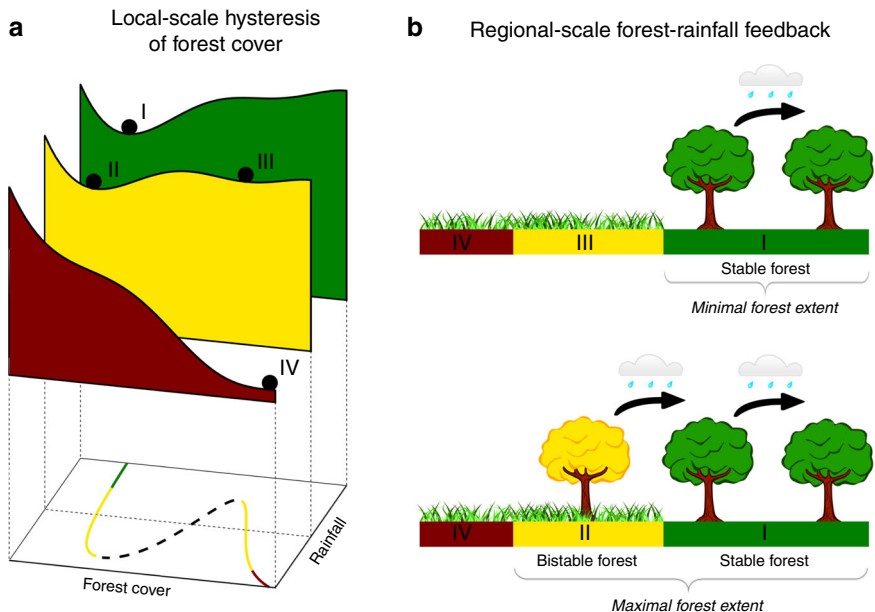

**Fig. 1 Local-scale hysteresis of forest cover and its interaction with the regional forest–rainfall feedback. a** A stability landscape of forest cover against rainfall levels. At high rainfall levels, high forest cover is uni-stable (I; green), called 'stable forest' throughout this paper. At intermediate rainfall levels, high forest cover (II) and low forest cover (III; nonforest) are bistable states (yellow). At low rainfall levels, only the nonforested state can exist (IV; red). **b** The regional forest–rainfall feedback amplifies hysteresis: minimal forest extent includes only stable forests (green), thereby lacking the rainfall enhancement by bistable forests; maximal forest extent includes forests that are bistable (yellow), which then contribute to downwind rainfall levels and may stabilize forests on those locations.

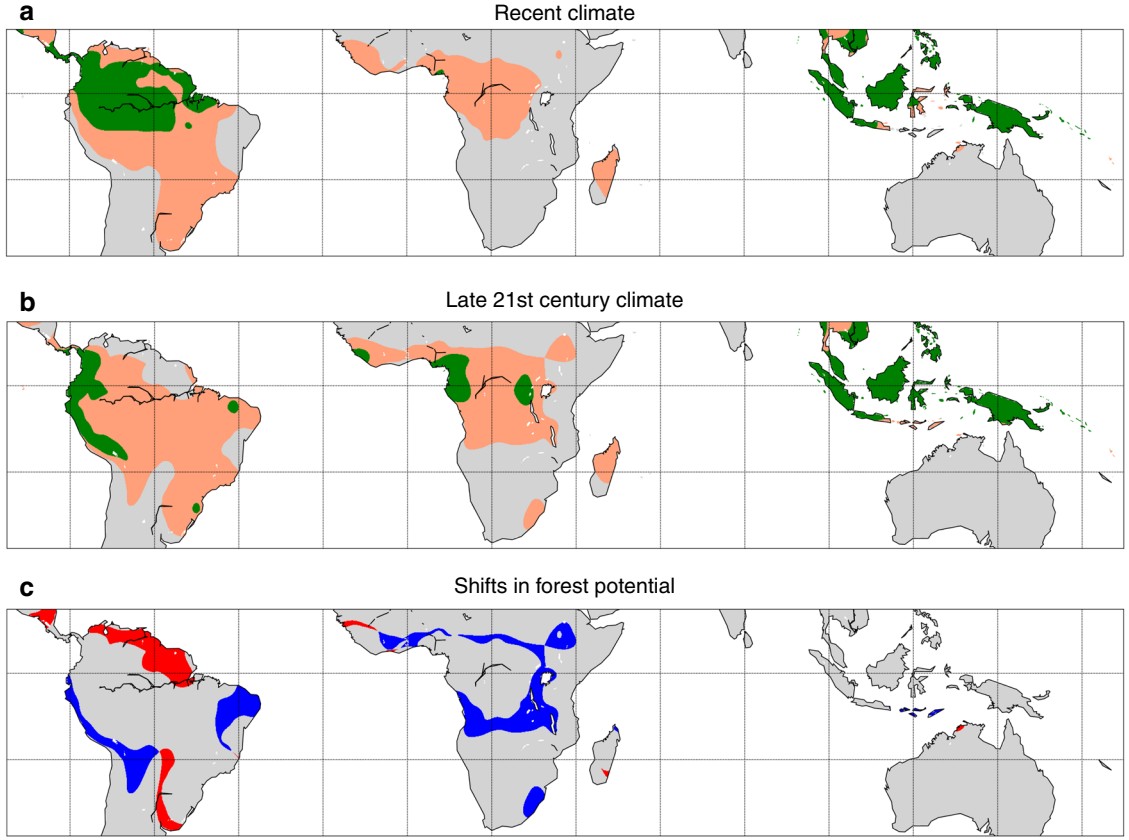

**Fig. 2 Changing hysteresis of forest cover in the tropics during the 21st century. a** Minimal (green) and maximal (beige) forest distributions under recent climate (2003−2014). **b** Minimal (green) and maximal (beige) forest distributions under the late 21st century climate (2071−2100). **c** Shifts in forest potential between the recent and late 21st century climates. Red areas are stable forest under the recent climate, but cross the tipping point into the nonforested rainfall range under the late 21st century climate; blue areas are too dry for forest under the recent climate, but cross the tipping point to the stable forest–rainfall range under the late 21st century climate. Note that these estimates are conservative in the sense that a rainfall change from the stable to the bistable range is assumed to have no effect on forest extent. For associated changes in rainfall, see Supplementary Figs. 6 and 7.

Australasia between 1550–1950 mm per year (Supplementary Figs. 1–5). In this paper, forests within these ranges are called 'bistable forests'. At rainfall levels above these ranges, forest cover is uni-stable—simply 'stable' from here on—meaning that we assume that forests always recover from natural disturbances. In South America, this applies to 4.93 million km² of current forest in the Amazon (Supplementary Fig. 5). Especially the central and northern Amazon contains stable forest, whereas the southern Amazon contains bistable forest. In Africa, only a small fraction (150,000 km²) of the Congo forest is stable, implying that the Congo forest is almost entirely bistable. In Australasia, 2.12 million km² forest, located in southeast Asia, is stable (Supplementary Fig. 5). These results highlight that especially in the Congo and the southern Amazon, disturbances such as fires can trigger tipping points, even without accounting for the forest–rainfall feedback[5].

Next, we use atmospheric moisture tracking of forest evapotranspiration to determine the effects of the forest–rainfall feedback. We simulate rainfall with forest cover removed and determine the minimal extent of forest cover (i.e. only the 'green forests' of Fig. 1) under these conditions. We iterate this procedure where, at each iteration, rainfall levels and forest distributions are updated depending on the forest–rainfall interactions. A minimum of 4.83 million km² of Amazon forest (60% of present extent) eventually recovered after complete deforestation, whereas 5.87 million km² of forest (72% of present extent) would recover if rainfall levels remain static with changing forest cover. In Africa, only 22,000 km² of forest recovered (1% of

present extent), relative to 120,000 km² under static rainfall levels (3% of present extent; compare Fig. 2a and Supplementary Fig. 13). In Australasia, 3.87 million km² forest area recovered (157% of present extent), almost the same as under static rainfall levels (158%; Fig. 2a).

Similar to the experiment to determine minimal forest extent, we simulate rainfall in case of full forest cover and determine the maximal extent of forest cover (i.e. retaining both the 'yellow' and 'green forests' of Fig. 1). In this experiment, forest-induced moisture recycling caused rainfall levels to be higher than they are in reality (Supplementary Figs. 8–10). We thus estimate that up to 12.26 million km² of forest can exist in tropical South America (156% of present extent), compared to 12.23 million km² under static rainfall levels (151% of present extent). In Africa, forest area stabilized at 5.35 million km² (140% of present extent), compared to 4.79 million km² under static rainfall levels (126% of present extent). Finally, in Australasia forest area stabilized at 4.57 million km² (185% of present extent), almost the same as under static rainfall levels (4.56 million km² or 185% of present extent). With these results for minimal and maximal forest extent with and without forest-induced moisture recycling, we can quantify by how much forest hysteresis (defined as the difference in forest cover area between the two extremes) is underestimated when the forest–rainfall feedback is unaccounted for. For South America, we find an increase in estimated forest hysteresis due to the forest–rainfall feedback of 22% to 7.79 million km², in Africa by 14% to 5.33 million km², and in Australasia by 2% to 0.69 million km² (Fig. 2a; Supplementary

Fig. 13a). Note that these numbers result from both increased maximal and decreased minimal forest extent under rainfall levels adjusted for the effects of forest cover relative to these extents under static rainfall levels.

We tested the sensitivity of hysteresis to a number of uncertain variables (see 'Methods'). We find that hysteresis is relatively insensitive to the share of evapotranspiration that is contributed by forest cover, although in Africa maximal forest extent increases visibly with forest evapotranspiration (Supplementary Fig. 18). Hysteresis is more sensitive to the values of bifurcation points, where higher values lead to larger estimated forest extents, and especially in Africa to smaller hysteresis (Supplementary Fig. 19). Recent research has shown that the greatest source of uncertainty in the atmospheric moisture tracking scheme is the rate of vertical mixing of atmospheric moisture[16]. We find that mixing rate has a small effect on hysteresis, but that higher atmospheric mixing tends to narrow the range between minimal and maximal forest extents (Supplementary Fig. 20), as stronger mixing causes forest evapotranspiration to rain down more locally[16].

Apart from mean annual rainfall, also other climatic variables, including rainfall variability, affect forest distributions and resilience regionally[17–19], while variations in soils and topography, and different biogeochemical functioning of forests, may affect them at local scales[20,21]. Therefore, it can be expected that the response of forests to climatic changes is more heterogeneous than assumed here. Based on maximum climatological water deficit (MCWD, a measure of dry season intensity; see 'Methods'), all tropical forests are estimated to be bistable (Supplementary Figs. 11 and 12). However, forest cover distributions suggest that forests are stable at sufficiently high mean annual rainfall levels even with some level of seasonality[5,19]. If we would classify landscapes as bistable if either mean rainfall or MCWD predicts bistability, we would obtain larger estimates of hysteresis than if we consider mean rainfall alone, which may lead to overestimation of hysteresis. We decided to adopt a conservative approach in estimating hysteresis by considering mean annual rainfall as the defining parameter for tropical forest stability. This agrees with findings that photosynthesis in the tropics is maintained year-round where rainfall levels exceed 2000 mm yr$^{-1}$[22]. Although rainfall seasonality has important effects on forest and savanna distributions and transitions, those effects occur generally within the mean annual rainfall levels that define the broad-scale hysteresis of tropical forests[5,8].

**Contrasting patterns under climate change**. We use rainfall projections under climate change to assess how the stability of tropical forests may change by the end of the century. We recognize that applying present statistical relations between forest distributions and mean annual rainfall to an average of a set of rainfall projections is a first-order approach. It disregards other important factors such as temperature change and changes in rainfall variability. Further, we do not account for tree adaptations, for example regarding water-use efficiency due to increasing $CO_2$ concentration or changes in carbon allocation by trees as a result of changing stress. However, it may provide some useful insights in the pattern and magnitude of the changes in forest stability that could result from its hysteresis behaviour under climate change. Therefore, we take the mean annual rainfall from the severe SSP5-8.5 scenario in seven CMIP6 model runs for the late 21st century (2071–2100; Supplementary Fig. 4) and thus assess the range of potential forest distributions across the tropics under climate change.

Global climate change will affect the hysteresis of tropical forests by the end of the 21st century. Notably, we find a large reduction of 66% to 1.66 million km$^2$ in minimal forest area for South America. The maximal forest area decreases by much less, namely by 4% to 12.15 million km$^2$. Although the area of maximal forest is hardly affected, its distribution is much more affected: an area of 1.91 million km$^2$ changes from unsuitable to suitable (i.e. either stable or bistable) for forest, whereas an area of 2.37 million km$^2$ changes from suitable to unsuitable. For Africa, we find a reversed pattern from that in South America: the minimal forest area increases by three orders of magnitude to 1.15 million km$^2$, and the maximal forest area by 54% to 8.26 million km$^2$. For Australasia, we find an increase of 9% in minimal forest to 4.24 million km$^2$, and an increase of 3% in maximal forest area to 4.72 million km$^2$ (Fig. 2).

In areas where climate change causes drying, some currently forested areas may cross a tipping point to a nonforested state, whereas in areas where climate change causes wetting, some currently nonforested areas may cross the reverse tipping point (Supplementary Figs. 1–3). We find that the former type of transition occurs mainly in South America, where 1.45 million km$^2$ of forest, located mainly in the northern Amazon, may cross a tipping point due to global climate change. In Africa only 3000 km$^2$ and in Australasia only 1000 km$^2$ of forest may cross this tipping point. The area that may undergo the reverse transition is more equally distributed across the continents: 660,000 km$^2$ in South America, 300,000 km$^2$ in Africa, and 310,000 km$^2$ in Australasia (Supplementary Fig. 14).

We assessed the how estimated hysteresis is affected by the choice of CMIP6 model. We find that this sensitivity is considerable, where rainfall levels and both the minimal and maximal forest extents can vary largely among model runs (Supplementary Figs. 21 and 22).

**Forest hysteresis effects on rainfall**. Forests play an important role in the hydrological cycle across the tropics, but their exact contribution, and that of forest hysteresis in particular, remains uncertain[23]. Using our simulations (of atmospheric moisture tracking of forest evapotranspiration) starting from a fully forested continent versus a nonforested continent, we can estimate the potential influence of forest hysteresis on the hydrological cycles on the different continents (Fig. 3). Current annual rainfall across tropical South America is on average 1700 mm per year. Upon starting simulations without any forest, it stabilized at 1600 mm per year. In this case, the forest precipitation recycling ratio (FPRR, which we define as the percentage of continental rainfall from forests) is 8%. Upon starting from a fully forested continent, it stabilized at 1790 mm per year, with a FPRR of 19%. For climate change we estimate an average annual rainfall of 1390 mm per year (FPRR = 1%) for the minimally stable area of forest and 1670 mm per year (FPRR = 11%) for the maximally stable area of forest (with 1580 mm per year without a change in forest; Fig. 3). Current average annual rainfall in tropical Africa is 990 mm per year. At minimal forest extent rainfall stabilized at 940 mm per year (FPRR = 0%) and at maximal extent at 1020 mm per year (FPRR = 10%). Climate change is projected to increase average rainfall levels in Africa to 1170 mm per year. At minimal forest extent we estimate a level of 1130 mm per year (FPRR = 0%) and at maximal forest extent 1220 mm per year (FPRR = 5%). In Australasia, forest hysteresis has a negligible effect on average rainfall levels, ranging between 1170 mm per year (FPRR = 1%) at minimal forest extent and 1180 mm per year (FPRR = 2%) at maximal forest extent. Climate change increases average rainfall to 1500 mm per year (FPRR = 0%) at minimal and maximal forest extent (Fig. 3). The small effect of forest in our Australasian study area can be explained by the discontinuous land area of Indonesia reducing the potential for terrestrial moisture recycling.

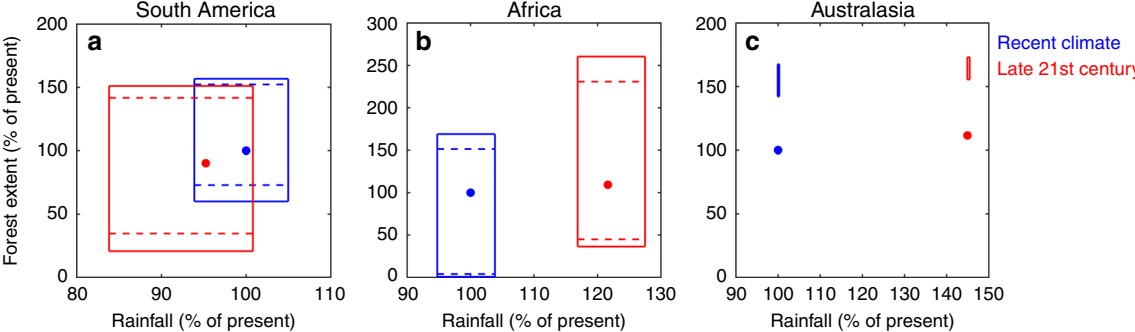

**Fig. 3 The ranges of the intensity of the hydrological cycle and total forest cover under recent and late 21st century climate.** Both the intensity of the hydrological cycle and total forest cover are given as percentage of their present value. The ranges that are delineated by solid lines result from the interaction of local-scale hysteresis of forest cover with regional-scale forest-induced moisture recycling, with those for the recent climate (2003−2014) given in red and those for the late 21st century climate (2071−2100) given in blue. The ranges that are delineated by dashed lines result from local-scale hysteresis only and do not account for regional-scale forest-induced moisture recycling (i.e. they assume 'static rainfall'). Note that the static climates by definition imply no ranges along the x-axis, but these ranges are kept for display purposes. Red dots indicate recent total annual rainfall and forest cover; blue dots indicate late 21st-century total rainfall and forest cover with no other factors than climate change considered. **a** South America; **b** Africa; and **c** Australasia. For spatial patterns in the ranges in rainfall levels, see Supplementary Figs. 6 and 7.

We predict a widening of the range of possible rainfall values resulting from forest hysteresis over the course of the 21st century. For South America, this is mostly due to a decrease in minimal forest area resulting from reduced advection under the applied climate-change scenario. For Africa, maximal forest area increases more strongly than minimal area in absolute terms, but minimal forest area increases more strongly in relative terms. In Australasia, climate change causes a large increase in rainfall levels, but the contribution of forest hysteresis to its possible range is negligible by comparison. Note that these results account for the differences in moisture recycling between minimal and maximal natural forest extent only. Additional effects of forest change on rainfall, such as through altered convection resulting from albedo changes, may also be substantial[24], and the active removal of stable forest could press the forest–rainfall system beyond the ranges given in Fig. 3.

## Discussion

Reforestation and afforestation in the tropics have been proposed as effective climate-change mitigation measures[25–31]. Given that the estimated forest potential includes natural grasslands and savannas[30], studies analysing the potential of afforestation implicitly acknowledge hysteresis in forest cover. However, such analyses do not account for changing potential forest distributions due to the rainfall effects of afforestation itself or their interactions with global climate change[32]. By accounting for these factors, our analysis sheds more light on the forest potential across the tropics[33], though it is important to note that afforesting natural grasslands and savannas may neither be a feasible nor desirable climate-change mitigation measure[30,34], and a number of other considerations, including biodiversity, would need to be accounted for ref. [33].

The existence of hysteresis due to local-scale feedbacks already implies that a multitude of tropical forest distributions are possible. As expected from theory[35], the regional-scale forest–rainfall feedback expands the range of possible distributions of forests, albeit to different extents on the different continents. This has implications for our understanding of the role of tropical forests in the Earth system. Whether the Amazon in particular is an important global 'tipping element' in the Earth system is a question of great scientific and societal interest[36,37]. Despite our incomplete understanding of Amazon tipping, it is generally considered to be true that the forest's role in the hydrological cycle is so large that deforestation and/or climate change may

trigger a tipping point[2,36–38]. More recently, the possibility of fire-induced tipping has also been suggested[5,6]. Although fire occurs at a local scale, a considerable portion of the Amazon would be susceptible to this kind of tipping; by accounting for the feedbacks at both local and regional scales, it becomes more likely that the Amazon is a tipping element. Although under the current climate a majority of the Amazon forest still appears resilient to disturbance (also see ref. [39]), we show that this resilience may deteriorate as a result of redistributions of rainfall due to global climate change. We further argue that the Congo rainforest should also be considered a tipping element. Because our results indicate that forest cover in the Congo is bistable, but that global climate change may enhance forest resilience, we suggest that deforestation has a potentially larger effect on its possible tipping than global climate change. Our results, however, do not indicate that the southeast Asian rainforests are tipping elements in the Earth system. Still, maintaining the climate-regulating functioning of tropical forests requires their conservation globally[1,40].

We found that hysteresis is rather robust against a number of uncertain factors under the current climate, but that estimates can greatly vary for the late 21st century depending on the climate model. For our main results, we use a multi-model average, which may cancel out some extremes. However, these results should be interpreted with caution. Furthermore, we assumed that wind patterns remain the same under the future climate, although climate models indicate that both latitudinal and longitudinal moisture fluxes will increase (Supplementary Fig. 23).

Caution should be taken not to overgeneralize the functioning of tropical forests. However, our results highlight a fundamental property of Earth's tropical forests: that forest extent is only partially determined by the environment. The hysteresis of tropical forests emerging from cross-scale feedbacks illustrates how the interplay between local and global changes can have lasting effects on the Earth system.

## Methods

**Study area and period.** Our study area is the tropics between 15°N–35°S[6]. We divided the study area into three continents and studied them separately: South America, Africa, and Australasia. Australasia includes Australia and southeast Asia, but excludes southern India. Our results are generated on 0.25° spatial resolution. We classify a cell as forest if it contained at least 50% tree cover ('forest cover' in this manuscript) in 1999 according to the dataset from ref. [41]. The moisture recycling simulations were carried out for 2003–2014 ('recent climate'), for which a consistent set of input data was available (see also ref. [11]). 'Late 21st century' refers to 2071–2100.

**Local-scale forest hysteresis**. Previous research has shown that tropical forests may have local-scale tipping points at certain mean annual rainfall levels, but are also affected by the seasonality of that rainfall[5,6,8]. Local-scale tipping points for forest were determined using tree cover data following a method from ref. [6]. Using potential analysis[42], an empirical stability landscape (as in Fig. 1a) is constructed based on spatial distributions of tree cover against environmental variables such as mean annual rainfall for each continent separately. For each value of the environmental variable, the probability density of tree cover was determined using the MATLAB function ksdensity with a bandwidth of 5%. We applied Gaussian weights to the environmental variable with a standard deviation of 0.05 times the length of the axis of the environmental variable. Local maxima of the resulting probability density function are interpreted as stable states, where we ignored local maxima below a threshold value of 0.004. We used Landsat tree cover data for 2000 on 30 m resolution downloaded for every 0.01°[43]. We masked out human-used areas, water bodies, and bare ground using the ESA GlobCover land cover dataset for 2009 on 300 m resolution (values 11–30 and ≥190). From the resulting dataset we randomly sampled one million locations for each continent and used them to construct the stability landscapes[6] against mean annual rainfall and average MCWD. MCWD is the cumulative difference between evapotranspiration and rainfall using monthly averages of those fluxes calculated for each calendar year[44]. It is set to zero when monthly rainfall exceeds monthly evapotranspiration and becomes more negative with an increasing water deficit. Following ref. [11], for both mean annual rainfall and MCWD, we took monthly data from the GLDAS 2.0 dataset[45] for 1970–1999 so the 30-year period leading up to the land-cover sample (for the year 2000) was used.

**Forest evapotranspiration**. To estimate the fraction of evapotranspiration attributable to forest cover we used the large-scale hydrological model PCR-GLOBWB, run at 0.5° resolution[46]. Per grid cell, the model simulates evapotranspiration for a range of land-cover types. Here, we are specifically interested in the evapotranspiration of forests, or 'tall natural vegetation' in PCR-GLOBWB. Note that we here account for both forest transpiration and canopy interception evaporation instead of, as in ref. [11], only transpiration.

PCR-GLOBWB computes the water balance in two soil layers and a groundwater layer. Soil type, fractional area of saturated soil, and the spatiotemporal distribution of groundwater depth are accounted for (see refs. [46,47]). It includes six land-cover types, with spatially varying parameters[46]: tall and short natural vegetation, pasture, rainfed crops, and paddy and non-paddy irrigated crops. The model was forced with WATCH Forcing Data ERA-Interim precipitation, temperature, and reference potential evapotranspiration for 1979–2014[48]. We used monthly evapotranspiration output of PCR-GLOBWB, implying that we assume that forest component of evapotranspiration remains equal within each month. For detailed model descriptions and validation, we refer to earlier studies[11,46,49,50].

**Atmospheric moisture tracking**. As an essential step in estimating the forest–rainfall feedback, we determined where the moisture from enhanced evapotranspiration precipitates again by using atmospheric moisture tracking. The method for atmospheric moisture tracking resembles that in ref. [11]. Apart from the expansion of the study area to the entire tropics, a notable difference is that we here used ERA5 reanalysis data rather than ERA-Interim, meaning that the simulations were based on finer resolution input data (i.e. on 0.25° instead of 0.75° for spatial resolution, and 1 h instead of 3 h for temporal resolution). ERA5 has better performance than ERA-Interim regarding wind fields and rainfall, especially in the tropics[51–53]. Below we summarize the method (see also ref. [11]).

We used a Lagrangian method of moisture tracking that is based on previous studies[11,54–56] that track parcels of evaporated moisture forward through the atmosphere to their subsequent precipitation location. Moisture particles that enter the atmosphere are assigned a random location within the 0.25° grid cell and random starting height in the atmosphere scaled with the humidity profile, and their trajectories are then tracked through the atmosphere. The trajectories are forced with the three-dimensional ERA5 reanalysis estimates of wind speed and direction, which were linearly interpolated at every time step of 0.25 h. Water particles in the atmosphere have an equal probability of raining out, regardless of vertical position. Rainfall $A$ (mm per time step) at location $x,y$ and time $t$ that has evaporated from any location of release in any cell is ref. [56]

$$A_{x,y,t} = P_{x,y,t} \frac{W_{\text{parcel},t} E_{\text{source},t}}{\text{TPW}_{x,y,t}}, \qquad (1)$$

where $P$ is rainfall in mm per time step, $W_{\text{parcel}}$ is the water in the tracked parcel in mm, $E_{\text{source}}$ is its fraction of water that evapotranspired from the source, and TPW is the precipitable water in the atmospheric water column in mm. Every time step, the amount of water in the parcel is updated based on evapotranspiration ET into the parcel and rainfall $P$ from it:

$$W_{\text{parcel},t} = W_{\text{parcel},t-1} + (\text{ET}_{x,y,t} - P_{x,y,t}) \frac{W_{\text{parcel},t-1}}{\text{TPW}_{x,y,t}}. \qquad (2)$$

The fraction of water in the parcel that has evapotranspired from the source then becomes

$$E_{\text{source},t} = \frac{E_{\text{source},t-1} W_{\text{parcel},t-1} - A_{x,y,t}}{W_{\text{parcel},t}}. \qquad (3)$$

Thus, the amount of water that was tracked from the source location decreases with precipitation along its trajectory. Parcels were followed until either less than 5% of its original amount was left in the atmosphere, or the tracking time was 30 days. Any moisture remaining in the parcel when the trajectories end is assumed to rain out over non-land areas, thus not contributing to our analysis. We analysed each continent separately for reasons of computability. However, small moisture flows between forests in different continents can be expected, as has been simulated for flows from Africa to the Amazon[57]. Over all land points, ERA5 hourly evapotranspiration is linearly interpolated to every 0.25-h time step. The moisture flow $m_{ij}$ in mm per month linking evapotranspiration in cell $i$ to rainfall in cell $j$ where $[x,y] \epsilon j$ over the course of a given month becomes

$$m_{ij} = \sum_{t=0}^{\text{month}} A_{j,t} \cdot \frac{\text{ET}_{i,t}}{W_{i,t}}, \qquad (4)$$

where $\text{ET}_{i,t}$ is the evapotranspiration in mm per time step, and $W_{i,t}$ is the tracked amount of water from source cell $i$ at time step $t$.

At continental scales, evaporated moisture can rain down and re-evaporate multiple times. This also means that forest evapotranspiration can enhance rainfall multiple times. We accounted for this 'cascading moisture recycling' following refs. [11,14], in which the rainfall attributed to an upwind forest source is subsequently tracked upon re-evaporation. After six re-evapotranspiration cycles, cascading moisture recycling has decreased to practically zero[11]. Therefore, following ref. [11], seven iterations of cascading moisture recycling were performed.

**Hysteresis experiments**. We determined the hysteresis of tropical forests through a series of iterative runs; each one started either from a fully forested continent or from a fully deforested continent. We simulated the hypothetical mean annual rainfall levels across the (tropical part of the) continent given this initial condition, that is, rainfall without any forest evapotranspiration or rainfall including evapotranspiration from an entirely forested continent. Next, based on the empirical bifurcation diagrams (i.e. nonforest, bistable forests, and stable forests in each continent are determined based on the bistability ranges shown in Supplementary Figs. 1−3), we determined either the minimal distribution of tropical forest (in case of a no-forest initial condition, based on the higher end of the bistability range) or the maximal distribution (in case of a fully forested initial condition, based on the lower end of the bistability range) at these rainfall levels. Thus, in the simulations with an empty initial condition, only stable forests (green in Fig. 1) would regrow; in those with a full initial condition, both stable and bistable forests (green and yellow in Fig. 1) would disappear. Because the resulting new distribution of forest would generate different levels of rainfall, the procedure was repeated with the respective forest distribution as initial condition. This occurred until rainfall levels had (practically) stabilized between iterative runs (up to three iterations).

We assumed that no other vegetation type replaces the forest in order to show the theoretically possible distributions of tropical forests. This may lead to an overestimation of the actual effects of forest on rainfall, especially if forests would be replaced by highly transpiring crops[58]. Furthermore, land-cover changes will alter wind patterns and therefore the expected coupling between forests through evapotranspiration and rainfall[59]. Fossil fuel emissions not only alter the climate, but the emitted $CO_2$ also fertilizes plants. This increases trees' water-use efficiency, reducing their water demand, but it also increases biomass production[60]. The effects of this $CO_2$ fertilization on the water cycle might be small[61], but its net effects on tropical forest hysteresis remains uncertain.

For display of Fig. 2, the resolution of rainfall values was increased by a factor of 2, to 0.125° and smoothed using a two-dimensional Gaussian kernel with a standard deviation of 0.5°.

**Climate-change scenario**. As the estimate of late 21st-century rainfall conditions, we used the rainfall output from the SSP5-8.5 scenario simulations by seven available CMIP6 models[62]. These models are BCC-CSM2-MR, CanESM5, CNRM-CM6-1, CNRM-ESM2, IPSL-CM6A-LR, MRI-ESM2.0, and UKESM1.0-LL. We took the mean across the model outputs for the annual rainfall values for 2071–2100. The scenario is considered a severe climate-change scenario. Because the moisture tracking model is forced with atmospheric reanalysis data, we assumed that (forest-induced) moisture flows in the scenario are the same as in the period of our atmospheric simulations (2003–2014).

We assumed that a tipping point from a forested to a nonforested state occurs when mean annual rainfall in a forested cell (forest cover ≥ 50%) is currently (2003–2014) above the lower tipping point (Supplementary Figs. 1–3), but is reduced to below the lower tipping point in the climate-change scenario. Similarly, a tipping point from a nonforested to a forested state occurs when mean annual rainfall in a nonforested cell (forest cover < 50%) is currently (2003–2014) below

the upper tipping point (Supplementary Figs. 1–3), but is increased to above the upper tipping point in the climate-change scenario.

To explore whether the CMIP6 models project a change in moisture transport for the late 21st century, we compared the vertically integrated eastward and northward moisture fluxes (in kg m$^{-1}$ s$^{-1}$) for 35°S–35°N for 2015–2020, which is the start of the simulation runs, and 2095−2100, the end of the runs. We did this for the same seven models and SSP as mentioned above.

**Validation and sensitivity analyses**. We conducted a number of additional analyses regarding model validation and uncertainties. We compared our evapotranspiration product GLDAS to estimates from FLUXCOM. Instead of using climate forcing data, FLUXCOM merges energy flux measurements from FLUXNET eddy covariance towers with remote sensing[63]. Thus, it provides an independent as possible comparison with GLDAS. Over all, the two products agree well with a concordance correlation $r^2 = 0.69$ across the tropics (Supplementary Fig. 15). This correspondence is lower when we consider forested areas only ($r^2 = 0.26$; Supplementary Fig. 16). Especially at relatively low values of monthly evapotranspiration they differ, where FLUXCOM tends to produce higher estimates of (forest) evapotranspiration than GLDAS. Positive and negative differences exist throughout the tropics, but especially in Africa, FLUXCOM estimates higher evaporation levels than GLDAS (Supplementary Fig. 17). Underestimations of evaporation by GLDAS would imply that changes in forest cover may have larger effects than we currently account for, but systematic bias in flux measurement data might also be responsible[63].

We assess the sensitivity of forest hysteresis on each continent to a number of variables. For these sensitivity analyses we performed our atmospheric simulations for 2003 only. We did this for: (1) the share that forest cover contributes to evapotranspiration, using 80, 90, 100, 110, and 120% of the estimated levels used in the main analyses. (2) The values of the bifurcation points, where we simultaneously changed both the lower and upper bifurcation point by −200, −100, 0, 100, and 200 mm per year. (3) The mixing strength of atmospheric moisture along the vertical moisture column. This was shown to be the most important source of uncertainty in Lagrangian atmospheric moisture tracking[16]. Here, we applied three levels of atmospheric mixing: low, in which moisture gets assigned a new random vertical location every 120 h; medium, used in the main analyses, in which moisture gets assigned a new location every 24 h; and high, where mixing occurs every hour. These specific analyses were done on 0.5° instead of 0.25°. (4) The CMIP6 climate model, where we estimated the hysteresis for each of the used models separately.

All data analyses were carried out in MATLAB R2019a. Figure 2 was made using Matplotlib 2.2.5.

**Reporting summary**. Further information on research design is available in the Nature Research Reporting Summary linked to this article.

## Data availability

The Landsat tree cover data are available at https://e4ftl01.cr.usgs.gov/MEASURES/GFCC30TC.003/. The PCR-GLOBWB hydrological model experiment was forced with WATCH ERA-Interim data available for download at ftp://ftp.iiasa.ac.at/. Further forcing data of the model are available for download at https://zenodo.org/record/1045339#.XzZlejVcJhF. The moisture tracking model used ERA5 data available for download at https://www.ecmwf.int/en/forecasts/datasets/reanalysis-datasets/era5 and GLDAS2 data available for download at https://disc.sci.gsfc.nasa.gov/datasets?keywords=GLDAS. FLUXCOM data can be downloaded from http://fluxcom.org/EF-Download/. ESA GlobCover data can be downloaded at http://due.esrin.esa.int/page_globcover.php. CMIP6 model output as downloaded from https://esgf-node.llnl.gov/projects/cmip6/. The data for Fig. 2 are available as Supplementary Data 1. For further requests, please contact the corresponding author.

## Code availability

The codes for the PCR-GLOBWB model are available at https://github.com/UU-Hydro/PCR-GLOBWB_model. The codes for the moisture tracking model are available at https://github.com/ObbeTuinenberg/UTrack-atmospheric-moisture. All further codes are available from the corresponding author upon request.

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

## Acknowledgements
We thank Chandrakant Singh for helpful discussions. A.S., I.F., L.W.-E., and J.R. acknowledge support from the European Research Council project Earth Resilience in the Anthropocene (743080 ERA). L.W.-E. acknowledges support from the Swedish Research Council Formas (2018-02345 and 2019-01220). O.A.T. acknowledges support from the research program Innovational Research Incentives Scheme 016.veni.171.019 by the Netherlands Organisation for Scientific Research (NWO).

## Author contributions
A.S. conceived the study. A.S., E.H.v.N., and O.A.T. designed the study. A.S., J.H.C.B., and O.A.T. carried out the study. A.S., I.F., L.W.-E., S.C.D., E.H.v.N., J.R., and O.A.T. interpreted the results. A.S. wrote the paper with contributions from I.F., L.W.-E., J.H.C.B., S.C.D., E.H.v.N., and O.A.T.

## Funding

## Competing interests
The authors declare no competing interests.
