## [Peer Review File · Nature Communications]

Reviewers' comments:

Reviewer #1 (Remarks to the Author):

The study explores the role of tropical forests in controlling their resilience to deforestation and response to climate change. The authors take a novel approach linking a range of modelling tools. The overall findings of the paper are important and influence the way we understand forest – rainfall interactions. The conclusions will be of interest to a wide range of researchers and policy makers. However, a number of issues require clarification before publication could be considered.

Major comments

A major comment is around the ability to evaluate or test the model results against real world observations. This study is an interesting model experiment, but it would be much more convincing if there was a stronger observational constraint. Can any elements of the analysis be tested with observations?

Another weakness is related to the model prediction of precipitation which appears to be based on available water for precipitation. This is particularly relevant when studying future climates. I think the authors need to provide some evidence that their approach is valid under future climate scenarios.

Minor comments

Figure 2. It would be useful to have a plot of simulated change in rainfall between the two periods. Does the change in forest extent in (c) match the change in rainfall.

Line 76-79. How does this approach account of deforested regions where tree cover is artificially low?

Line 142. Do you also exclude any impacts of rising CO₂ concentrations and changes in water-use efficiency.

Line 223. This study includes numerous errors that have been outlined in a number of critical comments. These errors means the conclusions of the paper are misleading. I would suggest providing a different reference to support this statement.

Line 260. Why do you exclude tropical forests between 23N and 15N?

Line 270. What do you mean by "high resolution forest data"? What dataset did you use? What is the resolution of the climate data used for this analysis? If the rainfall data is relatively coarse (0.25 degrees), what is the benefit of conducting this analysis at high resolution?

Line 283. Is there an issue around consistency of the input datasets used in the different modelling tools. I.e., GLDAS versus ERA-5.

Line 398. I'm not sure understand this statement. Does this mean that you do not account for changes in atmospheric circulation in a future climate? This seems like a major simplification that requires further justification.

Reviewer #2 (Remarks to the Author):

OVERALL ASSESSMENT:

The paper by Staal and co-authors presents an analysis of the hysteresis in forest cover throughout the tropics. They have used the geographic distribution of forests at the end of the 20th century and mean annual precipitation amounts during that period to estimate levels of rainfall where both forested and unforested ecosystems are both present, with these areas deemed 'bi-stable'. Using this stability landscape, they then estimated the maximal and minimal extent of cover under current and future climates. A particularly interesting analysis undertaken here is an evaluation of how maximal and minimal forest would be changed if climatological feedbacks are considered. Overall the paper is well written and clear, though there are a number of broad and specific points we need to be addressed as described below

BROAD CONCERNS

Much of the abstract is about rainfall-forest feedbacks and moisture recycling. This is an interesting part of the analysis that the authors have undertaken. However only a small portion of the text is focused on the rainfall-forest feedbacks. For instance, in the introduction, "we find that forest-rainfall feedback expands the range of possible forest distributions especially ". But how much of this is the hysteresis effect and how much is the rainfall forest feedback. No figures in the main text or SI compare distributions in the static and dynamic rainfall models. Improving the characterization of the effects of this mechanics would add to this paper.

Throughout this paper, a number of oversimplifications were made, particularly with respect to why different forest covers can occur at the same annual precipitation totals. Various other factors, such as rainfall seasonality and intensity, as well as soil properties are disregarded here, even though the authors own analysis demonstrates that the MCWD (a measure of dry season intensity) was clearly demonstrated to influence forest cover. In fact, this is a core logical fallacy of this paper. On one hand the technique of creating "stability landscapes" is useful for identifying bi-stability (as in the case of rainfall) but on the other hand the same technique isn't useful (as in the case of the MCWD) because "it is known that forests are stable at sufficiently high mean annual rainfall levels" (with the quote from L119 of the text). This acknowledges that the authors approach can produce spurious identification of bi-stable locations, so how/why are we to believe that the rainfall bi-stabilities' are the valid estimates of bi-stable locations.

A second major concern here is the role of uncertainty in this approach. Many of the datasets and methods here contain significant uncertainty. One reason CMIP includes multiple models is because it is unclear at this time which model is best, and the inter-model spread describes to some degree our uncertainty in future predictions. The same is true for precipitation estimates and land surface models for the modern era. However, there is no analysis of how potential errors and uncertainty in forest cover, bi-stable regions, model rainfall, etc. will influence the conclusions drawn, or why they were not considered. Given the simplified nature of the model of forest cover (i.e. it is solely based on mean annual precipitation) some effort should be made to convince the readers that this approach can accurately model forest distributions observed presently.

Also, moisture tracking is described in the methods but not in the paper main body. Why/where is this analysis done and what did you learn from it? Besides within the methods, the word 'tracking' only appears in the main body of the manuscript to state that it was done.

SPECIFIC CONCERNS

L16: I don't think this paper includes an analysis of "local-scale tipping points". Please rephrase.
L19: You do not show in the main body any of the moisture tracking results.

L20: Clarify that this is the geographic distribution, not PDF of forest covers.

L77: The landscape is what is bi-stable, not the forest. Another more fitting word choice would be the forest cover. With both forest and non-forest possible states. Same issues in line 80. Saying the forest is stable as a savanna is contradictory.

L90: The rainfall and forest's are not interacting at each time step. The forests are determining rainfall within the model. But the Rainfall only determines the forest cover outside of the model.

L90-97: Can you also express these numbers as fractions of the forest there. These differences may be pretty small as a percent of total area, or not?. Also add some of these to Figure 3.

L110: You don't find an increase in hysteresis. You find an increase in forest due to hysteresis.

L111: Is this with the static or dynamically estimated climates?

L113: Beyond resilience, a number of papers have shown that climate seasonality, rainfall intensity, soil properties, and other factors determine the geographic distribution of where forest are found.

L119: What is the basis of this statement?

L120: I fail to see how this is the 'conservative' approach. I would think that the conservative approach would be to consider both possible mechanism, in the event that multiple mechanisms are possibly affecting the outcome. You could have conducted this entire analysis with the MCWD stability landscapes for instance. How would this change your conclusions?

L173: Only hysteresis effects on rainfall are presented here, your model outputs other components of the hydrologic cycle, so either present these as well or change your section header to precipitation, since that is all you discuss.

L249: What is the basis of this statement.

L304: While this is interesting, I didn't see anywhere in the main text where this work is discussed. Either remove it from the methods or discuss it in the text.

Figure 3: Can you add on to this plot the levels of maximal and minimal forest under the static climate?

-Stephen Good

Reviewers' comments:

Reviewer #1 (Remarks to the Author):

The study explores the role of tropical forests in controlling their resilience to deforestation and response to climate change. The authors take a novel approach linking a range of modelling tools. The overall findings of the paper are important and influence the way we understand forest – rainfall interactions. The conclusions will be of interest to a wide range of researchers and policy makers. However, a number of issues require clarification before publication could be considered.

Major comments

A major comment is around the ability to evaluate or test the model results against real world observations. This study is an interesting model experiment, but it would be much more convincing if there was a stronger observational constraint. Can any elements of the analysis be tested with observations?

We thank the reviewer for this comment, which helps us to strengthen the evidence basis for our study. While some analytical part are actually already fully based on observations, other parts have different observational constraints. We identified the following elements of the analyses:

1. *local-scale hysteresis data and analysis* – the analysis (Hirota et al. 2011) already entirely relies on observation-based data (Sexton et al. 2013).
2. *forest evaporation data* - based on the reviewer's suggestion, we now added a comparison between GLDAS and the observation-based evapotranspiration dataset FLUXCOM to the revised manuscript. The PCR-GLOWB model has been evaluated in the past, and validated with good results against observed discharge data.
3. *atmospheric moisture recycling data and method* – the meteorological data from ERA5 used for moisture recycling is observation-based, and the tracking method has been evaluated against other methods and observations in past studies.
4. *climate scenarios* – we use CMIP6 model outputs that are evaluated against observations and that have been shown to reproduce very well observed large-scale mean surface temperature and precipitation patterns, with no known systematic errors in our study regions.

Below, we address each of the four above points in more detail.

Important to realize is that the final model results cannot be directly validated against observations. The reason that complete comparison cannot be carried out is because, by definition, we explore potential dynamics for tropical forests that are beyond the realm of observation: what are theoretically possible minimal and maximal possible extents of tropical forest, and how they can change in case of the most extreme future climate change scenarios?

1. Local-scale hysteresis

Our analysis of local-scale hysteresis (without the forest-rainfall feedback) depends entirely on observation-based data. We use remote sensing products of tree cover against mean annual rainfall to estimate hysteresis. There has been some discussion in the literature about how reliable MODIS-based tree cover products are for the purpose of constructing stability landscapes. Some bias in the remote sensing product was shown (Hanan et al. 2014), although this is much lower than the actual signal of bimodality on which our analysis is based (Staver and Hansen 2015).

Verification of the reliability of tree cover data for constructing stability landscapes came when the same analysis was done for both tree cover and canopy height data (Xu et al. 2016; Xu et al. 2018). This showed consistency between the two approaches: the bistability ranges as inferred from tree cover are the same as those inferred from canopy height, as bimodality in tree cover coincided with bimodality in canopy height. Some high-canopy and low tree cover areas exist, which indicate the presence of degraded forests rather than natural savannas. However, as said, this did not lead to a change in outcome from this analysis (Xu et al. 2018).

2. Forest evapotranspiration

As an additional analysis for this revision, we now compare the evapotranspiration that we have used, GLDAS, to a different observation-based dataset. To select this independent dataset, we took different considerations into account: 1) the scale of the data should be similar to allow for a comparison that allows for a clear interpretation; 2) the dataset should not be shown to work badly in the tropics; 3) the dataset should not be too similar in its design to GLDAS.

Given consideration 1, we discarded direct flux tower measurements. Such ground measurements may be valuable for validation of new gridded evaporation products, but for our purpose and our study scope, a gridded product that has already been comprehensively validated is preferred. Given consideration 2 we discarded MOD16 (Mu, Zhao, and Running 2013). While MOD16 has to our knowledge not been consistently evaluated in the tropical rainforests, it has low correlation with flux tower measurements in a number of land-use types (Souza et al. 2019), including temporal forests and sub-tropical woody savanna (Ramoelo et al. 2014). Moreover, evaluation of the Penman-Monteith based algorithm that underlies the remote sensing product MOD16 also indicated that MOD16 evapotranspiration may be underestimated in the tropics (Miralles et al. 2016). We further considered the three observation-based gridded evapotranspiration based datasets GLEAM, PML, and FLUXCOM. The former two use satellite input data as much as possible in a minimalistic process model and the latter is based on machine learning techniques. Given consideration 3 we chose the FLUXCOM remote sensing product, as it differs substantially from the method behind GLDAS, while GLEAM and PML do have similarities with GLDAS. Instead of using climate forcing data, the dataset merges energy flux measurements from FLUXNET eddy covariance towers with remote sensing (Jung et al. 2019).

We compared monthly GLDAS and FLUXCOM on a monthly and 0.25° basis (2003-2014) for each continent and the tropics combined. We did this separately for all data points and for only the forested areas. In addition, we map the difference in mean annual evaporation between the two datasets.

As can be seen from the figures below (added to the supplement as Figs. S15-S17), monthly evapotranspiration from GLDAS and FLUXCOM correspond well over-all: across the tropics, their linear r^2 (relative to the 1:1 line) is 0.69. Considering only forested areas, this correspondence is lower: with a linear r^2 of 0.26, there is medium correspondence between the two datasets. Especially at relatively low values of monthly evapotranspiration they differ, where FLUXCOM tends to produce higher estimates of (forest) evapotranspiration than GLDAS.

From the map of the difference between the two datasets it can be seen that areas of positive and negative difference exist throughout the tropics, but that especially in Africa, FLUXCOM estimates higher evapotranspiration levels than GLDAS. Underestimations of evapotranspiration by GLDAS would imply that changes in forest cover may have larger effects than we currently account for. However, as the sensitivity analyses of forest hysteresis to evapotranspiration changes presented in response to

comments from reviewer 2 show, sensitivity of hysteresis to evapotranspiration is relatively low. In addition, Jung et al. (2019) hypothesize that the higher carbon uptake in FLUXCOM in comparison to modelled data in the tropics could be due to systematic bias in flux measurement data. They don't mention possible bias in the modelled data that they compare FLUXCOM to. However, Mueller et al. (2013) show that GLDAS is on the lower side with their estimates.

We added these figures as Figs. S15-S17, described in the Methods in lines 466-479 in the new section "Validation and sensitivity analyses".

Monthly evaporation (2003-2014) from FLUXCOM versus GLDAS (all data points). The r² refers to the concordance correlation coefficient (correspondence along the 1:1 line).

Monthly evaporation (2003-2014) from FLUXCOM versus GLDAS (only forested areas). The r^2 refers to the concordance correlation coefficient (correspondence along the 1:1 line).

Mean annual evapotranspiration (2003-2014) from FLUXCOM minus that from GLDAS.

The PCRaster Global Water Balance model, used to estimate the forest (fraction of) evapotranspiration, is described in detail by (Sutanudjaja et al. 2018). They provide a model validation of discharge compared to data from the Global Runoff Data Centre (GRDC). Discharge is modelled well in monsoon-dominated areas, which cover a large part of the tropics. They do find that model results in first-world regions (e.g. US, Europe) compare better to observations, but this is related to the meteorological

forcing being of better quality in such areas due to better / more observations. In Africa, model deficiencies arise in the Niger area due to groundwater processes and underestimation of evaporation over the delta regions. We acknowledge that no model validation or uncertainty testing has been done with respect to evapotranspiration, but we assume that when discharge is modelled well, so is evapotranspiration.

3. Atmospheric moisture tracking

Our atmospheric moisture tracking is carried out based on meteorological data from ERA5, which is numerical weather model outputs adjusted using data-assimilation from remotely sensed data, and therefore can be considered to be partly observation-based. Previous moisture tracking comparisons between the reanalysis data of ERA-Interim and MERRA shows that performance are similar in general, but that ERA-Interim performs better than MERRA in South America due to an overestimation of Atlantic ocean evaporation contribution to land precipitation in MERRA (Keys et al. 2014).

Direct observations of moisture recycling pathways for validation of model-based tracking cannot be obtained. Indirect observation of moisture recycling pathways include remote-sensing leaf-area index (Spracklen, Arnold, and Taylor 2012; Spracklen and Garcia-Carreras 2015) and isotope studies (e.g. Yoshimura 2015; Galewsky et al. 2016). Neither of these methods, however, generate datasets that can be used for direct validations, and the interpretation of isotope measurements also requires modelling and contains substantial uncertainty. Isotopic data are also minimal; the recent project MUSICA (Schneider et al. 2017) may offer a way to validate moisture tracking models, but is originally designed to validate moisture pathways in atmospheric models rather than moisture tracking with regard to specific terrestrial sources. We are not aware of any attempts to make use of these data for moisture tracking validation.

Due to comparisons between different types of moisture tracking models in previous studies, we can be confident that the moisture tracking models are fairly reliable over large areas and time scales (as in this study). For example, van der Ent et al. (2013) showed that tagging scheme based on Eulerian tracking, Eulerian tracking on model coordinates, and Lagrangian tracking on Eulerian coordinates (this study) produced fairly similar global patterns. Moisture recycling sources over the Amazon and Congo appear to be fairly consistent among different studies (Gimeno et al. 2020).

There are some uncertainties related to the moisture tracking, which possibly affect the current study. These include (1) forcing data with possibly too few atmospheric layers in regions with strong horizontal shear, (2) insufficient amount of tracer parcels released in Lagrangian models, and (3) uncertain vertical mixing assumptions. In this study, the Lagrangian model took into account 25 atmospheric layers, released 10 traced parcels per mm of ET at surface level and tracked these for 30 days, or until 99% of the moisture was allocated. As is demonstrated in Tuinenburg and Staal (2020), the number of parcels is adequate over these monthly timescales. Furthermore, our trajectory model is forced with a high density of vertical layers in the lower troposphere, where the vast majority of atmospheric moisture is located. Therefore, we argue that the main uncertainty in the moisture tracking is due to the vertical mixing assumptions, as also demonstrated in Tuinenburg and Staal (2020). As we further discuss in response to reviewer 2, we have done a number of sensitivity experiments with respect to the vertical mixing during tracking. In the standard model, every moisture parcel has a probability to be vertically mixed along the moisture profile. This probability is such that this mixing happens once per day for each parcel in average. In the sensitivity experiments, we have changed the probability such that the mixing happens once per hour (strong mixing), once per 24 hours (medium mixing) and once per 120 hours (weak mixing). The results of these mixing experiments on the overall analysis show low sensitivity to

mixing speed. This is presented in the supplementary analysis as Fig. S20; it is also pasted along with results of other sensitivity analyses below, in response to reviewer 2.

4. Climate scenarios

For the future projections of the climatic conditions we rely on the CMIP6 model scenario outputs. Its predecessor CMIP5 has been used in the IPCC 5th assessment report (IPCC AR5 <https://www.ipcc.ch>) and shown to mostly replicate features of the global climate system (see IPCC AR5 Chapter 9 and figure 9.24). Climate experts therefore have very high confidence in the validity of the model outputs as they are also permanently corrected for errors (Schmidt, Shindell, and Tsigaridis 2014).

With the new modelling round in CMIP6 the range of climate variables and processes that have been evaluated has greatly expanded (with among others increasing resolutions of 25 x 25 km, altitudinal corrections and cloud formations and -feedbacks, etc.) but is also more rigorously evaluated (<https://pcmdi.llnl.gov/CMIP6/>). Differences between models and observations are regularly quantified using 'performance metrics' (see e.g. <https://www.climate-lab-book.ac.uk/comparing-cmip5-observations/>). The evaluation criteria of model in the CMIP6 cover mean current climate, historical climate change, variability on multiple time scales and regional modes of variability.

In general, it has been shown that the models reproduce very well observed large-scale mean surface temperature and precipitation patterns. Systematic errors are mainly found for high altitude areas, in some North Atlantic regions, and over areas of ocean upwelling near the equator, as here characteristic local climatic characteristics have strong impacts that cannot be covered by the models. Therefore, especially on regional scales (sub-continental and smaller), the confidence in model capability to simulate the climate is lower than for the larger scales.

Another weakness is related to the model prediction of precipitation which appears to be based on available water for precipitation. This is particularly relevant when studying future climates. I think the authors need to provide some evidence that their approach is valid under future climate scenarios.

The big uncertainty regarding the validity of our approach relates to the correspondence of future moisture recycling patterns to those of the present. Our moisture recycling analyses are based on atmospheric reanalysis data, so they run a posteriori. The climate scenarios are applied offline and are provided the same moisture recycling patterns as in the present. Thus, the extension of our analysis into the late 21st century must be seen as a crude first step in understanding the role of the forest-rainfall feedback in shaping tropical forest hysteresis into the future. As a first-order check for validity of our approach under future climate scenarios, we obtained the average vertically integrated latitudinal and longitudinal moisture flows from the CMIP6 models that we used. Here, we compare 2015-2020, which is the start of the scenario runs, with 2095-2100, which is the end. This result is newly included as supplementary Fig. S23 and pasted below. Blue lines indicate the mean for 2015-2020 \pm one standard deviation, and the red lines 2095-2100 \pm one standard deviation. It can be seen that the CMIP6 models project larger moisture fluxes by the end of the 21st century. We did not account for this, but acknowledge this limitation in lines 301-303. We also expanded the Methods accordingly.

Minor comments

Figure 2. It would be useful to have a plot of simulated change in rainfall between the two periods. Does the change in forest extent in (c) match the change in rainfall.

Figures S4B-C show the mean annual rainfall for 2003-2014 and 2071-2100. Figures S6A-D show the mean annual rainfall levels for 2003-2014 under both minimal and maximal forest cover, and for 2071-2100 under both minimal and maximal forest cover. We understand that for comparison with Figure 2C, it is useful to have a plot that specifically shows the simulated change in rainfall between 2003-2014 and 2071-2100 under the most extreme scenarios, that is, 2003-2014 under minimal cover and 2071-2100 under maximal cover. Therefore, we added a new Figure S7 that shows this. We refer to this new figure in the captions of Figs. 2 and 3.

Difference in mean annual rainfall levels in mm yr^{-1} between the recent climate (2003-2014) at minimal forest and the SSP5-8.5 scenario for 2071-2100 at maximal forest extent. The color scale was cut off at the high end at 3000 mm yr^{-1} .

Line 76-79. How does this approach account of deforested regions where tree cover is artificially low?

We accounted for human land-use changes in estimating bistability by first masking out human-used areas according to the 2009 ESA GlobCover dataset (as well as water bodies and bare ground; values 11-30 and ≥ 190 in the dataset). This was mentioned in the Methods, but not in the main text. We added this statement to the main text (line 76-77): “Forest cover distributions (excluding human-used areas, water bodies and bare ground; see Methods) ...”.

Line 142. Do you also exclude any impacts of rising CO2 concentrations and changes in water-use efficiency.

Unfortunately, we do not. We agree that this is important to mention here, so we added that, and more (lines 171-174): "It disregards other important factors such as temperature change and changes in rainfall variability. Further, we do not account for tree adaptations, for example regarding water-use efficiency due to increasing CO₂ concentration or changes in carbon allocation by trees as a result of changing stress."

Line 223. This study includes numerous errors that have been outlined in a number of critical comments. These errors means the conclusions of the paper are misleading. I would suggest providing a different reference to support this statement.

The study by (Bastin et al. 2019) is indeed rather controversial. To reflect the current debate better, we decided to change the sentence to "Reforestation and afforestation in the tropics are currently discussed as a climate change mitigation measure", where we refer to a number of the responses to Bastin et al. (2019) and other relevant work as well (Friedlingstein et al. 2019; Grainger et al. 2019; Griscom et al. 2020; Lewis et al. 2019; Skidmore et al. 2019; Veldman et al. 2019).

Furthermore, because of the controversy, we already referred in (the former) line 220 to the study by (Brancalion et al. 2019), which, to our knowledge, is not controversial. It specifically targets the tropics and takes into account biodiversity and social considerations. Mainly because of the recent discussions sparked by the Bastin et al. paper, we also added the statement that "... though it is important to note that afforesting natural grasslands and savannas may neither be a feasible nor desirable climate change mitigation measure, and a number of other considerations, including biodiversity, would need to be accounted for."

Line 260. Why do you exclude tropical forests between 23N and 15N?

We are aware that different studies adopt different definitions of the tropics. The main reason we excluded that area is that it is consistent with the study domain of (Hirota et al. 2011), which is an important building block for our study: this influential study first outlined the methodology for inferring local-scale hysteresis of forest cover in the tropics. In its slipstream, a number of studies on forest-savanna bistability have adopted the same study area (Holmgren et al. 2013; Staal et al. 2016; Staal, van Nes, et al. 2018; Van Nes et al. 2014, 2018). We chose for consistency with the results presented in those papers and therefore refer to Hirota et al. where we define our study area (line 312).

Line 270. What do you mean by "high resolution forest data"? What dataset did you use? What is the resolution of the climate data used for this analysis? If the rainfall data is relatively coarse (0.25 degrees), what is the benefit of conducting this analysis at high resolution?

The details of the data (Landsat tree cover data for 2000 on 30 m resolution) are given a few lines below. To avoid confusion, we removed the early reference to "high resolution" in (the former) line 270.

The benefit of using high-resolution tree cover data - despite the coarser resolution of rainfall data - is that tree cover varies on much smaller spatial scales than rainfall. It is inherent in the bistability of tree cover that under the same rainfall conditions (e.g., within one coarser pixel for rainfall) tree cover may be in either of alternative stable states. If we would, for instance, average tree cover to the scale of the rainfall data, we would lose the local-scale bimodality and with that the statistical signature of bistability. Such details can be found in Hirota et al. (2011), to which we refer here.

Line 283. Is there an issue around consistency of the input datasets used in the different modelling tools. I.e., GLDAS versus ERA-5.

As in previous work (Staal, Tuinenburg, et al. 2018), we used the GLDAS evaporation products, because that is a model product that is focused on simulating realistic land surface fluxes, whereas the atmospheric reanalysis products prioritize minimizing the atmospheric errors. Moreover, GLDAS provides a more detailed ET separation into bare soil, transpiration, skin reservoir evaporation, etc. Due to the data assimilation (most dominantly near-surface humidity and temperature) in the reanalysis, moisture may be added to or removed from the land surface (Tuinenburg and de Vries 2017), which may affect the land surface fluxes. This set-up means that there may be an inconsistency in the moisture budget (GLDAS ET mismatching with the ERA5 humidity changes). It should be noted that this mismatch is also already present in the ERA5 reanalysis itself, due to its inability to conserve moisture. Most importantly, we believe that this mismatch will not affect our results much. Compared to the other fluxes and the atmospheric flow variability, ET is a flux that is relatively constant in time. Therefore, we expect the GLDAS ET and ERA5 ET to be well temporally correlated. This means that we are releasing moisture into the atmosphere at comparable moments, and effectively sampling the atmospheric flow regimes the same for GLDAS ET as would have been the case for ERA5 ET. We added the consistency with (Staal, Tuinenburg, et al. 2018) in line 334.

Line 398. I'm not sure understand this statement. Does this mean that you do not account for changes in atmospheric circulation in a future climate? This seems like a major simplification that requires further justification.

That is correct. Moisture tracking models at present cannot account for changes in future winds as they are based on atmospheric reanalysis data. It is a research frontier in this field to study future changes in moisture recycling and it is outside the scope of our current work. We agree it is important to acknowledge and justify that clearly in lines 301-303.

Reviewer #2 (Remarks to the Author):

OVERALL ASSESSMENT:

The paper by Staal and co-authors presents an analysis of the hysteresis in forest cover throughout the tropics. They have used the geographic distribution of forests at the end of the 20th century and mean annual precipitation amounts during that period to estimate levels of rainfall where both forested and unforested ecosystems are both present, with these area's deemed 'bi-stable'. Using this stability landscape, they then estimated the maximal and minimal extent of cover under current and future climates. A particularly interesting analysis undertaken here is an evaluation of how maximal and minimal forest would be changed if climatological feedbacks are considered. Overall the paper is well written and clear, though there are a number of broad and specific points we need to be addressed as described below

Thank you for these encouraging words.

For this revision we implemented one minor adjustment in the method regarding the calculation of (un)stable forest area. In the previous version, the calculation for the area of minimally stable forest were affected by the current forest cover distributions. Thus, for example, if a forest patch has 90% cover but is predicted to be stable/resilient, it counted as (area × 90%) forest cover. Also, forest cover in areas that were predicted to be unstable for 2003-2014 conditions (which occurred occasionally) was retained in the estimate of minimally stable forest cover. Similarly, and more importantly, non-forested areas that were predicted to be uni-stable forest were not classified as stable. This means that deforested areas under high rainfall levels (the stable forest domain) were not included in the estimated area of stable forest. In the new version, we calculated the minimal and maximal forest extents regardless of present land cover. We believe that this adjustment better reflects the aims of our study and ensures internal consistency. It resulted in some quantitative changes in the forest extent along the vertical in Figure 3 and to slight changes in the maps in Figure 2. The only qualitative change that resulted from it is that minimal forest area in Australasia is now above 100% of its present extent. This is due to the fact that in many areas of predicted stable/resilient forest cover, observed forest cover is below 100%.

BROAD CONCERNS

Much of the abstract is about rainfall-forest feedbacks and moisture recycling. This is an interesting part of the analysis that the authors have undertaken. However only a small portion of the text is focused on the rainfall-forest feedbacks. For instance, in the introduction, “we find that forest-rainfall feedback expands the range of possible forest distributions especially “. But how much of this is the hysteresis effect and how much is the rainfall forest feedback. No figures in the main text or SI compare distributions in the static and dynamic rainfall models. Improving the characterization of the effects of this mechanics would add to this paper.

Throughout this paper, a number of oversimplifications were made, particularly with respect to why different forest covers can occur at the same annual precipitation totals. Various other factors, such as rainfall seasonality and intensity, as well as soil properties are disregarded here, even though the authors own analysis demonstrates that the MCWD (a measure of dry season intensity) was clearly demonstrated to influence forest cover. In fact, this is a core logical fallacy of this paper. On one hand the technique of creating “stability landscapes” is useful for identifying bi-stability (as in the case of rainfall) but on the other hand the same technique isn’t useful (as in the case of the MCWD) because “it is known that forests are stable at sufficiently high mean annual rainfall levels” (with the quote from L119 of the text). This acknowledges that the authors approach can produce spurious identification of bi-stable locations, so how/why are we to believe that the rainfall bi-stabilities’ are the valid estimates of bi-stable locations.

We understand this concern, but do not agree that it implies a logical fallacy of our work. We think that a misunderstanding arises from a difference in interpretation of the importance of MCWD as a control variable for forest cover bistability.

The technique of creating stability landscapes is useful only when the resilience of the response variable (here, forest cover) is affected by the control variable monotonically. If it is not, then that means that other factors are more important. If mentioned control variable is still used to estimate bistability, then, as the reviewer rightfully states, the approach can produce spurious identification of bistable locations.

In the figures below we plot the ratio of forested cells over nonforested cells along bins of mean rainfall and MCWD. (We used one million data points for each continent, the same as used in the manuscript to construct the bifurcation plots.) A higher ratio means that forests are relatively more stable. It can be clearly seen that mean rainfall relates fairly monotonically and consistently (i.e. across rainfall values and across continents) to forest stability, but that MCWD does not. That is the reason why the use of MCWD as control variable may result in mentioned spurious identifications, but mean rainfall does not.

South America

Africa

A second major concern here is the role of uncertainty in this approach. Many of the datasets and methods here contain significant uncertainty. One reason CMIP includes multiple models is because it is unclear at this time which model is best, and the inter-model spread describes to some degree our uncertainty in future predictions. The same is true for precipitation estimates and land surface models for the modern era. However, there is no analysis of how potential errors and uncertainty in forest cover, bi-stable regions, model rainfall, etc. will influence the conclusions drawn, or why they were not considered. Given the simplified nature of the model of forest cover (i.e. it is solely based on mean annual precipitation) some effort should be made to convince the readers that this approach can accurately model forest distributions observed presently.

Thank you for this suggestion. We have performed a set of analyses to test the sensitivity of forest hysteresis to a number of important uncertain variables. These variables are: forest evapotranspiration, forest cover bifurcation points, atmospheric mixing speed of moisture, and CMIP6 model. This resulted in a considerable number of additional runs with large computational demand, which is why we performed all sensitivity analyses for the atmospheric moisture flows of 2003 only. The figures for the relative effects on predicted forest hysteresis below.

It can be seen from the figures that the sensitivity of hysteresis to the uncertainty in forest evapotranspiration and atmospheric mixing speed is small. We find larger sensitivity to bifurcation points. A higher value for the lower of the two bifurcation points reduces maximal forest area and a higher value of the upper bifurcation point reduces minimal forest extent on each continent. For South America and Australasia, the same increase for both of the bifurcation points (thus maintaining the width of local-scale forest hysteresis) increases the continental-scale hysteresis, while for Africa, an increase in bifurcation points reduces continental-scale hysteresis.

The greatest sensitivity occurs for the late 21st-century climate related to the CMIP6 models. There is large spread around the hysteresis for multi-model average mean rainfall values (as in the main text). For Africa and Australasia, the lowest estimate for maximal forest extent is smaller than the highest estimate for minimal forest extent. We also mapped the uncertainties across CMIP6 models by showing mean annual precipitation across models, median annual precipitation, minimal precipitation, maximal

precipitation, maximal minus minimal precipitation and the coefficient of variation in mean precipitation across models. It can be seen from, for instance, maximal minus minimal precipitation, that the uncertainties occur on all continents.

We added these figures to the supplementary material (Figures S18-S21), include a discussion about these sensitivities to the main text (lines 126-135) and expanded the methods accordingly (lines 481-493).

Also, moisture tracking is described in the methods but not in the paper main body. Why/where is this analysis done and what did you learn from it? Besides within the methods, the word 'tracking' only appears in the main body of the manuscript to state that it was done.

Moisture tracking is the key method to estimate the regional forest-rainfall feedback. Without it, we cannot determine where forest evapotranspiration precipitates and thus map out forest hysteresis due to this effect. We start presenting the results including moisture tracking from the second paragraph of the Results & Discussion. To prevent any confusion, we now start off that paragraph with "Next, we use atmospheric moisture tracking of forest evapotranspiration to determine the effects of the forest rainfall feedback." (lines 93-94)

The most important lesson from these simulations is that "For South America, we find an increase in estimated forest hysteresis due to the forest-rainfall feedback of 22% to 7.79 million km², in Africa by 14% to 5.33 million km², and in Australasia by 2% to 0.69 million km² (Figs. 2a, S13a)." (lines 119-122)

SPECIFIC CONCERNS

L16: I don't think this paper includes an analysis of "local-scale tipping points". Please rephrase.

In this paper, we do provide original estimates of local-scale tipping points of forest cover, based on Landsat forest cover data at 30 m resolution (Figs. S1-S3). Tipping point is here defined as a saddle-node bifurcation point for forest cover against mean annual rainfall, consistent with commonly used definitions of tipping points (Lenton 2013; Van Nes et al. 2016). In other words, the tipping points delimit the hysteresis. We clarified this in the Introduction in lines 55-56, where we state that we base our analysis on "remote-sensing-based estimates of local hysteresis as delimited by local-scale tipping points".

We believe that "we determine the emergent hysteresis from local-scale tipping points and regional-scale forest-rainfall feedbacks" in the Abstract reflects the contents of our study, so we prefer to retain that wording.

L19: You do not show in the main body any of the moisture tracking results.

It is true that we did not present moisture tracking results in isolation; but of course, figures 2 and 3 and all hysteresis simulations depend on these moisture tracking simulations. In addition, we now calculated the forest precipitation recycling ratio under the different scenarios. We define the forest precipitation recycling ratio as the percentage of continental precipitation for which forests are responsible. For South America, we find that at minimal forest extent under the current climate this ratio is 8% and at maximal forest extent it is 19%. For Africa, the respective ratios are 0% and 10%. For Australasia they are 1% and 2%. Under severe climate change, minimal and maximal forest precipitation recycling ratios for South America are 1% and 11%, for Africa 0% and 6%, and for Australasia 0% and 0%. We provide these results in the main text in the section "Forest hysteresis effects on rainfall" (lines 209-261).

L20: Clarify that this is the geographic distribution, not PDF of forest covers.

We rephrased to "the geographic range of possible forest distributions".

L77: The landscape is what is bi-stable, not the forest. Another more fitting word choice would be the

forest cover. With both forest and non-forest possible states. Same issues in line 80. Saying the forest is stable as a savanna is contradictory.

We changed this section to: “Forest cover distributions (excluding human-used areas, water bodies and bare ground; see Methods) indicate that forest cover in South America is bistable between mean annual rainfall levels of 1,250–2,050 mm yr⁻¹; within this range, both forests and a savanna-like nonforested state are found. For Africa we find this bistability between 1,350–2,050 mm yr⁻¹, and in Australasia between 1,550–1,950 mm yr⁻¹ (Figs. S1–S5). In this paper, forests within these ranges are called ‘bistable forests’. At rainfall levels above these ranges, forest cover is uni-stable—simply ‘stable’ from here on—meaning that forests always recover from disturbances.”

L90: The rainfall and forest’s are not interacting at each time step. The forests are determining rainfall within the model. But the Rainfall only determines the forest cover outside of the model.

That is correct. We rephrased this section as follows (lines 93-97): “Next, we use atmospheric moisture tracking of forest evapotranspiration to determine the effects of the forest rainfall feedback. We simulate rainfall with forest cover removed and determine the minimal extent of forest cover (i.e. only the ‘green forests’ of Fig. 1) under these conditions. We iterate this procedure where at each iteration, rainfall levels and forest distributions are updated depending on the forest-rainfall interactions.”

We rephrased the beginning of the next paragraph (lines 106-108) as “Similar to the experiment to determine minimal forest extent, we simulate rainfall in case of full forest cover and determine the maximal extent of forest cover (i.e. retaining both the ‘yellow’ and ‘green forests’ of Fig. 1).”

L90-97: Can you also express these numbers as fractions of the forest there. These differences may be pretty small as a percent of total area, or not?. Also add some of these to Figure 3.

We included the fractions of the forest in this paragraph as well as the following one (lines 93-124). Figure 3 has percentages of current forest on the y axis. Depending on the continent this ranges between 1% to 158% fractional change of current forest.

L110: You don’t find an increase in hysteresis. You find an increase in forest due to hysteresis.

The effect of the forest-rainfall feedback on forest hysteresis works in two directions: on the one hand, it increases the maximal forest extent, as extensive forest partly creates the conditions enabling itself. On the other hand, at the minimal forest extent, rainfall would be lower than it is at present. Therefore, the forest-rainfall feedback also decreases the minimal forest extent relative to a situation with rainfall levels that are independent of forest cover. In other words, while it is true that the forest-rainfall feedback increases maximal forest extent, it also decreases minimal forest extent *relative to its estimate based on observed rainfall levels and forest cover distributions*. Thus, it expands *estimated* hysteresis.

We understand that this point was not clear enough in the text, so we added the following sentence (lines 122-124): “Note that these numbers result from both increased maximal and decreased minimal forest extent under rainfall levels adjusted for the effects of forest cover relative to these extents under static rainfall levels.”

Also, inspired by the last comment (below), we included the forest cover ranges under ‘static climates’ (i.e. in the absence of forest-rainfall feedback) in Figure 3. It can be seen that the hysteresis in this case is more narrow from both sides than when the forest-rainfall feedback is accounted for.

L111: Is this with the static or dynamically estimated climates?

We hope that the added sentence under the previous comment clarifies this.

L113: Beyond resilience, a number of papers have shown that climate seasonality, rainfall intensity, soil properties, and other factors determine the geographic distribution of where forest are found.

That is correct. To better reflect this we revised the text and added more references (lines 137-139): “Apart from mean annual rainfall, also other climatic variables, including rainfall variability, affect forest distributions and resilience regionally (Holmgren et al. 2013; X. Xu et al. 2018; Ciemer et al. 2019), while variations in soils and topography may affect them on local scales (Daskin, Aires, and Staver 2019; Flores et al. 2020).”

L119: What is the basis of this statement?

This could indeed have been written more clearly and carefully, as well as properly referenced. We revised this sentence to (lines 143-145): “However, forest cover distributions suggest that forests are stable at sufficiently high mean annual rainfall levels even with some level of seasonality (Ciemer et al. 2019; Staver, Archibald, and Levin 2011).”

L120: I fail to see how this is the ‘conservative’ approach. I would think that the conservative approach would be to consider both possible mechanism, in the event that multiple mechanisms are possibly affecting the outcome. You could have conducted this entire analysis with the MCWD stability landscapes for instance. How would this change your conclusions?

The reasoning behind our ‘conservative approach’ is the same as explained in response to the first broad concern. As shown in Fig. S12, if we use MCWD as control variable for forest stability, we end up predicting the entire tropics to be bistable. The reason is that MCWD by itself does not consistently predict forest resilience, as illustrated in the figures presented in response to the first concern.

As a further illustration of our reasoning, consider the following example. A forest has a mean annual rainfall of 3000 mm and a mean MCWD of 100 mm. Based on the one-dimensional stability landscapes against each of these independent variables, different conclusions can be drawn: based on mean rainfall, we would conclude that this forest is stable/resilient, because nonforested areas with 3000 mm annual rainfall are very rare, regardless of MCWD (Figs. S1-S3). Based on MCWD alone, we would conclude that forest cover is bistable under these conditions (Fig. S11). Thus, if we base our analysis on MCWD we would overestimate hysteresis. Importantly, this does not occur the other way around. Thus, mean rainfall is the dominant variable demarcating the bistability range although MCWD may still affect stability. It is in this sense that we adopt a conservative approach: if we would consider landscapes bistable if either mean rainfall or MCWD predicts bistability, we would obtain larger estimates of hysteresis, which, given the above, would result in overestimations of hysteresis. We expanded the explanations in the manuscript in lines 145-147.

L173: Only hysteresis effects on rainfall are presented here, your model outputs other components of

the hydrologic cycle, so either present these as well or change your section header to precipitation, since that is all you discuss.

Indeed, here we only discuss the rainfall component of the hydrological cycle, so we changed the section header to “Forest hysteresis effects on rainfall”.

L249: What is the basis of this statement.

The basis for this statement that “deforestation, not global climate change, could make [the Congo forest] tip” is 1) that we found that most of the Congo forest is bistable and therefore potentially sensitive to the deforestation-induced reductions in rainfall; and 2) the climate scenarios predict increases in mean rainfall in this region, increasing the forest’s resilience. However, we realize that this is a strong statement that should be weakened. We revised the respective sentences to (lines 290-293): “Because our results indicate that the Congo forest is bistable, but that global climate change may enhance its resilience, we suggest that deforestation has a potentially larger effect on its possible tipping than global climate change.”

L304: While this is interesting, I didn’t see anywhere in the main text where this work is discussed. Either remove it from the methods or discuss it in the text.

Atmospheric moisture tracking is an essential part of our analysis of the forest-rainfall feedback (Fig. 1b), so this section cannot be removed from the methods. Where we discuss results regarding the forest-rainfall feedback in the main text, we depend on the moisture tracking simulations. We tracked the forest-induced evapotranspiration from its source (location of evapotranspiration) to its sink (the location of precipitation) to quantify the extent to which forest cover enhances regional rainfall levels. This point was not so clear in the methods as this section was not linked to the previous section about forest evapotranspiration. Therefore, we added the following sentence at the start of this section (lines 355-357): “As an essential step in estimating the forest-rainfall feedback, we determined where the moisture from enhanced evapotranspiration precipitates again by using atmospheric moisture tracking.”

Figure 3: Can you add on to this plot the levels of maximal and minimal forest under the static climate?

That is a good idea. We added dashed lines representing the minimal and maximal forest extent for each continent and each climate period to Figure 3 and changed the caption accordingly.

-Stephen Good

References

- Bastin, Jean-Francois, Yelena Finegold, Claude Garcia, Danilo Mollicone, Marcelo Rezende, Devin Routh, Constantin M. Zohner, and Thomas W. Crowther. 2019. "The Global Tree Restoration Potential." *Science* 365(6448):76.
- Brancalion, Pedro H. S., Aidin Niamir, Eben Broadbent, Renato Crouzeilles, Felipe S. M. Barros, Angelica M. Almeyda Zambrano, Alessandro Baccini, James Aronson, Scott Goetz, J. Leighton Reid, Bernardo B. N. Strassburg, Sarah Wilson, and Robin L. Chazdon. 2019. "Global Restoration Opportunities in Tropical Rainforest Landscapes." *Science Advances* 5(7):eaav3223.
- Ciemer, Catrin, Niklas Boers, Marina Hirota, Jürgen Kurths, Finn Müller-Hansen, Rafael S. Oliveira, and Ricarda Winkelmann. 2019. "Higher Resilience to Climatic Disturbances in Tropical Vegetation Exposed to More Variable Rainfall." *Nature Geoscience* 12:174–79.
- Daskin, Joshua H., Filipe Aires, and A. Carla Staver. 2019. "Determinants of Tree Cover in Tropical Floodplains." *Proceedings of the Royal Society B: Biological Sciences* 286(1914):20191755.
- Flores, Bernardo M., Arie Staal, Catarina C. Jakovac, Marina Hirota, Milena Holmgren, and Rafael S. Oliveira. 2020. "Soil Erosion as a Resilience Drain in Disturbed Tropical Forests." *Plant and Soil*.
- Friedlingstein, Pierre, Myles Allen, Josep G. Canadell, Glen P. Peters, and Sonia I. Seneviratne. 2019. "Comment on 'The Global Tree Restoration Potential.'" *Science* 366(6463):eaay8060.
- Galewsky, Joseph, Hans Christian Steen-Larsen, Robert D. Field, John Worden, Camille Risi, and Matthias Schneider. 2016. "Stable Isotopes in Atmospheric Water Vapor and Applications to the Hydrologic Cycle." *Reviews of Geophysics* 54(4):809–65.
- Gimeno, Luis, Marta Vázquez, Jorge Eiras-Barca, Rogert Sorí, Milica Stojanovic, Iago Algarra, Raquel Nieto, Alexandre M. Ramos, Ana María Durán-Quesada, and Francina Dominguez. 2020. "Recent Progress on the Sources of Continental Precipitation as Revealed by Moisture Transport Analysis." *Earth-Science Reviews* 201:103070.
- Grainger, Alan, Louis R. Iverson, Gregg H. Marland, and Anantha Prasad. 2019. "Comment on 'The Global Tree Restoration Potential.'" *Science* 366(6463):eaay8334.
- Griscom, Bronson W., Jonah Busch, Susan C. Cook-Patton, Peter W. Ellis, Jason Funk, Sara M. Leavitt, Guy Lomax, Will R. Turner, Melissa Chapman, Jens Engelmann, Noel P. Gurwick, Emily Landis, Deborah Lawrence, Yadvinder Malhi, Lisa Schindler Murray, Diego Navarrete, Stephanie Roe, Sabrina Scull, Pete Smith, Charlotte Streck, Wayne S. Walker, and Thomas Worthington. 2020. "National Mitigation Potential from Natural Climate Solutions in the Tropics." *Philosophical Transactions of the Royal Society B: Biological Sciences* 375(1794):20190126.
- Hanan, Niall P., Andrew T. Tredennick, Lara Prihodko, Gabriela Bucini, and Justin Dohn. 2014. "Analysis of Stable States in Global Savannas: Is the CART Pulling the Horse?" *Global Ecology and Biogeography* 23(3):259–63.
- Hirota, Marina, Milena Holmgren, Egbert H. van Nes, and Marten Scheffer. 2011. "Global Resilience of Tropical Forest and Savanna to Critical Transitions." *Science* 334(6053):232–235.

- Holmgren, Milena, Marina Hirota, Egbert H. van Nes, and Marten Scheffer. 2013. "Effects of Interannual Climate Variability on Tropical Tree Cover." *Nature Climate Change* 3(8):755–758.
- Jung, M., C. Schwalm, M. Migliavacca, S. Walther, G. Camps-Valls, S. Koirala, P. Anthoni, S. Besnard, P. Bodesheim, N. Carvalhais, F. Chevallier, F. Gans, D. S. Groll, V. Haverd, K. Ichii, A. K. Jain, J. Liu, D. Lombardozzi, J. E. M. S. Nabel, J. A. Nelson, M. Pallandt, D. Papale, W. Peters, J. Pongratz, C. Rödenbeck, S. Sitch, G. Tramontana, U. Weber, M. Reichstein, P. Koehler, M. O’Sullivan, and A. Walker. 2019. "Scaling Carbon Fluxes from Eddy Covariance Sites to Globe: Synthesis and Evaluation of the FLUXCOM Approach." *Biogeosciences Discuss.* 2019:1–40.
- Jung, Martin, Sujana Koirala, Ulrich Weber, Kazuhito Ichii, Fabian Gans, Gustau Camps-Valls, Dario Papale, Christopher Schwalm, Gianluca Tramontana, and Markus Reichstein. 2019. "The FLUXCOM Ensemble of Global Land-Atmosphere Energy Fluxes." *Scientific Data* 6(1):74.
- Keys, Patrick W., EA Barnes, RJ van der Ent, and Line J. Gordon. 2014. "Variability of Moisture Recycling Using a Precipitationshed Framework." *Hydrology and Earth System Sciences* 18(10):3937–50.
- Lenton, Timothy M. 2013. "Environmental Tipping Points." *Annual Review of Environment and Resources* 38:1–29.
- Lewis, Simon L., Edward T. A. Mitchard, Colin Prentice, Mark Maslin, and Ben Poulter. 2019. "Comment on 'The Global Tree Restoration Potential.'" *Science* 366(6463):eaaz0388.
- Miralles, D. G., C. Jiménez, Martin Jung, D. Michel, Ali Ershadi, M. F. McCabe, Martin Hirschi, Brecht Martens, A. J. Dolman, J. B. Fisher, and O. Mu. 2016. "The WACMOS-ET Project-Part 2: Evaluation of Global Terrestrial Evaporation Data Sets." *Hydrology and Earth System Sciences* 20(2):823–42.
- Mu, Qiaozhen, Maosheng Zhao, and Steven W. Running. 2013. "MODIS Global Terrestrial Evapotranspiration (ET) Product (NASA MOD16A2/A3)." *Algorithm Theoretical Basis Document, Collection 5.*
- Mueller, Brigitte, Martin Hirschi, C. Jimenez, P. Ciais, P. A. Dirmeyer, A. J. Dolman, J. B. Fisher, Martin Jung, F. Ludwig, and F. Maignan. 2013. "Benchmark Products for Land Evapotranspiration: LandFlux-EVAL Multi-Data Set Synthesis." *Hydrology and Earth System Sciences* 17:3707–20.
- Ramoelo, Abel, Nobuhle Majozi, Renaud Mathieu, Nebo Jovanovic, Alecia Nickless, and Sebinasi Dziki. 2014. "Validation of Global Evapotranspiration Product (MOD16) Using Flux Tower Data in the African Savanna, South Africa." *Remote Sensing* 6(8):7406–23.
- Schmidt, Gavin A., Drew T. Shindell, and Kostas Tsigaridis. 2014. "Reconciling Warming Trends." *Nature Geoscience* 7(3):158–60.
- Schneider, M., C. Borger, A. Wiegeler, F. Hase, O. E. García, E. Sepúlveda, and M. Werner. 2017. "MUSICA MetOp/IASI {H₂O,ΔD} Pair Retrieval Simulations for Validating Tropospheric Moisture Pathways in Atmospheric Models." *Atmos. Meas. Tech.* 10(2):507–25.
- Sexton, Joseph O., Xiao-Peng Song, Min Feng, Praveen Noojipady, Anupam Anand, Chengquan Huang, Do-Hyung Kim, Kathrine M. Collins, Saurabh Channan, and Charlene DiMiceli. 2013. "Global, 30-

m Resolution Continuous Fields of Tree Cover: Landsat-Based Rescaling of MODIS Vegetation Continuous Fields with Lidar-Based Estimates of Error.” *International Journal of Digital Earth* 6(5):427–48.

Skidmore, Andrew K., Tiejun Wang, Kees de Bie, and Petter Pilesjö. 2019. “Comment on ‘The Global Tree Restoration Potential.’” *Science* 366(6469):eaaz0111.

Souza, Vanessa de Arruda, Débora Regina Roberti, Anderson Luis Ruhoff, Tamíres Zimmer, Daniela Santini Adamatti, Luis Gustavo G. de Gonçalves, Marcelo Bortoluzzi Diaz, Rita de Cássia Marques Alves, and Osvaldo LL de Moraes. 2019. “Evaluation of MOD16 Algorithm over Irrigated Rice Paddy Using Flux Tower Measurements in Southern Brazil.” *Water* 11(9):1911.

Spracklen, D. V., S. R. Arnold, and C. M. Taylor. 2012. “Observations of Increased Tropical Rainfall Preceded by Air Passage over Forests.” *Nature* 489(7415):282–285.

Spracklen, D. V., and Luis Garcia-Carreras. 2015. “The Impact of Amazonian Deforestation on Amazon Basin Rainfall.” *Geophysical Research Letters* 42(21):9546–9552.

Staal, Arie, Stefan C. Dekker, Chi Xu, and Egbert H. van Nes. 2016. “Bistability, Spatial Interaction, and the Distribution of Tropical Forests and Savannas.” *Ecosystems* 19(6):1080–1091.

Staal, Arie, Egbert H. van Nes, Stijn Hantson, Milena Holmgren, Stefan C. Dekker, Salvador Pueyo, Chi Xu, and Marten Scheffer. 2018. “Resilience of Tropical Tree Cover: The Roles of Climate, Fire, and Herbivory.” *Global Change Biology* 24(11):5096–5109.

Staal, Arie, Obbe A. Tuinenburg, Joyce H. C. Bosmans, Milena Holmgren, Egbert H. van Nes, Marten Scheffer, Delphine Clara Zemp, and Stefan C. Dekker. 2018. “Forest-Rainfall Cascades Buffer against Drought across the Amazon.” *Nature Climate Change* 8(6):539–43.

Staver, A. C., S. Archibald, and S. A. Levin. 2011. “The Global Extent and Determinants of Savanna and Forest as Alternative Biome States.” *Science* 334(6053):230–32.

Staver, A. Carla, and Matthew C. Hansen. 2015. “Analysis of Stable States in Global Savannas: Is the CART Pulling the Horse?—A Comment.” *Global Ecology and Biogeography* 24:985–87.

Sutanudjaja, E. H., R. van Beek, N. Wanders, Y. Wada, J. H. C. Bosmans, N. Drost, R. J. van der Ent, I. E. M. de Graaf, J. M. Hoch, K. de Jong, D. Karssenber, P. López López, S. Peş senteiner, O. Schmitz, M. W. Straatsma, E. Vannamete, D. Wisser, and M. F. P. Bierkens. 2018. “PCR-GLOBWB 2: A 5 Arcmin Global Hydrological and Water Resources Model.” *Geoscientific Model Development* 11(6):2429–2453.

Tuinenburg, O. A., and A. Staal. 2020. “Tracking the Global Flows of Atmospheric Moisture and Associated Uncertainties.” *Hydrology and Earth System Sciences* in press.

Tuinenburg, O. A., and J. P. R. de Vries. 2017. “Irrigation Patterns Resemble ERA-Interim Reanalysis Soil Moisture Additions.” *Geophysical Research Letters* 44(20):10,341–10,348.

- Van der Ent, R. J., O. A. Tuinenburg, H. R. Knoche, H. Kunstmann, and H. H. G. Savenije. 2013. "Should We Use a Simple or Complex Model for Moisture Recycling and Atmospheric Moisture Tracking?" *Hydrology and Earth System Sciences* 17(12):4869–84.
- Van Nes, Egbert H., Babak M. S. Arani, Arie Staal, Bregje van der Bolt, Bernardo M. Flores, Sebastian Bathiany, and Marten Scheffer. 2016. "What Do You Mean, 'Tipping Point'?" *Trends in Ecology & Evolution* 31(12):902–904.
- Van Nes, Egbert H., Marina Hirota, Milena Holmgren, and Marten Scheffer. 2014. "Tipping Points in Tropical Tree Cover: Linking Theory to Data." *Global Change Biology* 20(3):1016–1021.
- Van Nes, Egbert H., Arie Staal, Stijn Hantson, Milena Holmgren, Salvador Pueyo, Rafael E. Bernardi, Bernardo M. Flores, Chi Xu, and Marten Scheffer. 2018. "Fire Forbids Fifty-Fifty Forest." *PLoS ONE* 18(1):e0191027.
- Veldman, Joseph W., Julie C. Aleman, Swanni T. Alvarado, T. Michael Anderson, Sally Archibald, William J. Bond, Thomas W. Boutton, Nina Buchmann, Elise Buisson, Josep G. Canadell, Michele de Sá Dechoum, Milton H. Diaz-Toribio, Giselda Durigan, John J. Ewel, G. Wilson Fernandes, Alessandra Fidelis, Forrest Fleischman, Stephen P. Good, Daniel M. Griffith, Julia-Maria Hermann, William A. Hoffmann, Soizig Le Stradic, Caroline E. R. Lehmann, Gregory Mahy, Ashish N. Nerlekar, Jesse B. Nippert, Reed F. Noss, Colin P. Osborne, Gerhard E. Overbeck, Catherine L. Parr, Juli G. Pausas, R. Toby Pennington, Michael P. Perring, Francis E. Putz, Jayashree Ratnam, Mahesh Sankaran, Isabel B. Schmidt, Christine B. Schmitt, Fernando A. O. Silveira, A. Carla Staver, Nicola Stevens, Christopher J. Still, Caroline A. E. Strömberg, Vicky M. Temperton, J. Morgan Varner, and Nicholas P. Zaloumis. 2019. "Comment on 'The Global Tree Restoration Potential.'" *Science* 366(6463):eaay7976.
- Xu, Chi, Stijn Hantson, Milena Holmgren, Egbert H. van Nes, Arie Staal, and Marten Scheffer. 2016. "Remotely Sensed Canopy Height Reveals Three Pantropical Ecosystem States." *Ecology* 97(9):2518–2521.
- Xu, Chi, Arie Staal, Stijn Hantson, Milena Holmgren, Egbert H. van Nes, and Marten Scheffer. 2018. "Remotely Sensed Canopy Height Reveals Three Pantropical Ecosystem States: Reply." *Ecology* 99(1):235–37.
- Xu, Xiangtao, David Medvigy, Anna T. Trugman, Kaiyu Guan, Stephen P. Good, and Ignacio Rodriguez-Iturbe. 2018. "Tree Cover Shows Strong Sensitivity to Precipitation Variability across the Global Tropics." *Global Ecology and Biogeography* 27(4):450–60.
- Yoshimura, Kei. 2015. "Stable Water Isotopes in Climatology, Meteorology, and Hydrology: A Review." *Journal of the Meteorological Society of Japan. Ser. II* 93(5):513–33.

REVIEWER COMMENTS

Reviewer #1 (Remarks to the Author):

The authors have responded to all my comments and those of the other referee. The author responses are detailed and largely address my concerns or suitably describe why the comment is difficult to address. Revisions have been made to the manuscript and the revised manuscript is much improved. Overall, I think that the revised manuscript is suitable for publication.

Reviewer #3 (Remarks to the Author):

Thank you for your invitation to review this stimulating paper by Staal and collaborators. The paper proposes a simple, yet important, exercise to assess the potential effects of rainfall changes on the distribution of forests in a latitudinal band within the tropics. As a global exercise, the paper requires a great deal of assumptions that, eventually, can affect its results, particularly at the local-to-regional scales (the main focus of the paper). For instance, in South America, the interaction between the Andes and the Amazon is a fundamental aspect of the regional climate and determines the ecological, hydrological and biogeochemical dynamics of the Amazon. No differentiation between forest types can lead to potential overgeneralizations about the functioning of these important forests. Collectively, this global-scale exercise of local-to-regional forest-water interaction can suffer from accumulation of assumptions and biases from its definitions, data sources and models. Overlooking local-to-regional particularities, as well as fundamental biogeochemical aspects of forest structure and functions can limit the ability to conclude about forest resilience (and hysteresis). This accumulation of uncertainties leads to a potentially poor prediction capacity.

The concept of ecosystem hysteresis is very important and current, as world's ecosystems respond to more pressure from multiple disturbances. However, given that the mechanisms associated with forest function go beyond those explored here, perhaps predicting whole ecosystem hysteresis properties with only forest-rainfall interactions can be a little bold as forests are more than tree cover. I suggest making sure this is stated at the beginning.

Specific comments:

13. Very bold opening statement

30. What (quantitatively) is a rainfall level? There are more precise ways to define rainfall regimes

33. Perhaps what is "bimodal" is not the ecosystem per se but rather the (artificial) classification system. Most land cover classification systems use a quasi-arbitrary land/canopy cover threshold to differentiate a forest from a non-forest. However, more recent literature on ecotones has different approaches to define forest/non-forest transitions.

44. This is relative and perhaps only valid up to a certain point. See effects of recent (2005 and 2014) droughts in the Amazon

79. Most savannas in Northern South America (Venezuela and Colombia) receive more than 2000 mm year⁻¹ and they, naturally, are not forests. Perhaps this assumption needs to be validated.

84. This is a critical assumption: the only potential disturbance to forest stability is rain? "Forests always recover from disturbance" Is this true, particularly in the light of current changes in which multiple disturbances occur simultaneously?

98. 83 million km²? 87 million km²?

147. Is this really a conservative approach? I would argue that the potential to cross a tipping point (and therefore, overestimating bi-stability) is not conservative given the uncertain effects of climate change

151. As mentioned before, Forest-savanna transition in Northern South America does not follow the same set of rules.

213. Fully forested forest?

348. Any comment on the performance of ERA-Interim in the tropics (particularly in mountain regions)?

Reviewer #1 (Remarks to the Author):

The authors have responded to all my comments and those of the other referee. The author responses are detailed and largely address my concerns or suitably describe why the comment is difficult to address. Revisions have been made to the manuscript and the revised manuscript is much improved. Overall, I think that the revised manuscript is suitable for publication.

Thank you. We are happy that our responses and the revisions were satisfactory.

Reviewer #3 (Remarks to the Author):

Thank you for your invitation to review this stimulating paper by Staal and collaborators. The paper proposes a simple, yet important, exercise to assess the potential effects of rainfall changes on the distribution of forests in a latitudinal band within the tropics. As a global exercise, the paper requires a great deal of assumptions that, eventually can affect its results, particularly at the local-to-regional scales (the main focus of the paper). For instance, in South America, the interaction between the Andes and the Amazon is a fundamental aspect of the regional climate and determines the ecological, hydrological and biogeochemical dynamics of the Amazon. No differentiation between forest types can lead to potential overgeneralizations about the functioning of these important forests. Collectively, this global-scale exercise of local-to-regional forest-water interaction can suffer from accumulation of assumptions and biases from its definitions, data sources and models.

Overlooking local-to-regional particularities, as well as fundamental biogeochemical aspects of forest structure and functions can limit the ability to conclude about forest resilience (and hysteresis) This accumulation of uncertainties leads to a potentially poor prediction capacity.

Thank you for the encouraging words and critical reflection. We agree that many local to regional differences in forest function were not accounted for and that we rely on a number of assumptions. However, although assumptions and local uncertainties may limit the understanding on the regional resilience of forests, we address a general phenomenon (hysteresis) that creates even larger prediction uncertainties that concerns the forest as a whole and so far has not been investigated well. Moreover, to a certain extent we did account for regional hydroclimatic interactions and differences. The hydrological model PCR-GLOBWB resolves the forest cover-evapotranspiration relation at 0.5° resolution based on local land cover and climate, with spatially explicit parameterizations and thus differentiates among different types of forest. Regional interactions via forest-induced moisture recycling are resolved at 0.25° resolution. Local forest cover hysteresis against rainfall levels is resolved at continental scale. Thus, different forest cover thresholds between, for instance, the Andes

and Amazon are not accounted for. We added further words of caution at the end of the Discussion, in line 297: *“Caution should be taken not to overgeneralize the functioning of tropical forests.”*

Our simple approach to modelling forest hysteresis has the benefit of limiting the “accumulation of uncertainties” resulting from accumulation of model errors that comes with more parameters, assumptions, and complex model structures. Still, our analysis should be considered a first-order assessment (as also emphasized in lines 170-172) of tropical forest hysteresis due to combined local-scale hysteresis and regional forest-rainfall interactions, under current climatic conditions and even more so under late-21st-century conditions.

Including detailed biogeochemical aspects of forest structure and functions would go far beyond the scope of our study, but we agree that they may be important to tropical forest hysteresis. We acknowledge this limitation now in lines 140-142: *“... while variations in soils and topography, and different biogeochemical functioning of forests, may affect [forest distributions and resilience] at local scales.”*

The concept of ecosystem hysteresis is very important and current, as world’s ecosystems respond to more pressure from multiple disturbances. However, given that the mechanisms associated with forest function go beyond those explored here, perhaps predicting whole ecosystem hysteresis properties with only forest-rainfall interactions can be a little bold as forests are more than tree cover. I suggest making sure this is stated at the beginning.

We agree. Many factors affect forest distributions and hysteresis, not only rainfall. We added such a statement about these many factors in lines 42-43: *“... among the many factors that affect present forest extent is past forest extent; in other words, the system exhibits hysteresis”*. Next, we highlight the role of rainfall (lines 43-44): *“Moreover, the importance of past forest extent could be amplified by forest-rainfall interactions.”*

Specific comments:

13. Very bold opening statement

We are convinced that the statement that “tropical forests modify the conditions they depend on through feedbacks at different spatial scales” is justified. It is well-established that tropical forests contribute to evapotranspiration, which enhances rainfall regionally, although the extent to which (and where exactly) that occurs is surrounded with much more uncertainty. The fact that (tropical) forests depend on rainfall levels is, of course, clear. Regarding more local scales: in recent years evidence has been accumulating for a forest cover-fire feedback (Bowman et al., 2015; Staver et al., 2011; Van Nes et al., 2018). Additionally, other feedbacks such as between forest cover and soil fertility (Flores et al.,

2020) and how tropical forests mitigate warming through evaporative cooling (Bonan, 2008) are probably at play as well.

30. What (quantitatively) is a rainfall level? There are more precise ways to define rainfall regimes
By rainfall level we mean the amount of rainfall within a certain time frame and can thus be characterized at different temporal scales. At a range of these scales, forests may enhance the amount of rainfall. The average atmospheric residence time of transpired moisture is around nine days (Van der Ent et al., 2014), so a change in photosynthetic activity may result in rainfall changes at such short time scales (Spracklen et al., 2012). Furthermore, in the Amazon for instance, forests enhance rainfall levels especially at seasonal time scales (Staal et al., 2018). Of course, this rainfall enhancement is then also reflected in mean annual rainfall levels. Because the statement applies so generally, we chose the (indeed rather unspecified) “rainfall levels”. To avoid confusion, and consistent with our analyses, we now added “... *enhancing rainfall levels at seasonal to annual time scales*” (lines 30-31). We also added to the methods section (lines 314-315) the role of these time scales: “*Previous research has shown that tropical forests may have local-scale tipping points at certain mean annual rainfall levels, but are also affected by the seasonality of that rainfall.*”

33. Perhaps what is “bimodal” is not the ecosystem per se but rather the (artificial) classification system. Most land cover classification systems use a quasi-arbitrary land/canopy cover threshold to differentiate a forest from a non-forest. However, more recent literature on ecotones has different approaches to define forest/non-forest transitions.

That is a good point, however, the papers we refer to here (Hirota et al., 2011; Staver et al., 2011), as well as the potential analysis in our own paper, are based on continuous values of tree cover. In other words, it does not rely on any land cover classification. In this manuscript, we consistently refer to “forest cover” regardless of spatial scale, but specify the equivalence to tree cover as used in the above papers at the beginning of the introduction (lines 33-35): “... *the distribution of continuous values of tree cover (‘forest cover’ from here on) is distinctly bimodal. In other words, generally, either a fully covered forest or a sparsely covered nonforest (savanna or grassland) is found*”. We also added it at the beginning of the methods (line 309): “(‘forest cover’ in this manuscript)”. There has been discussion in the literature about potential biases in the underlying remote sensing data and the possibility of bimodality being an artefact of such biases (Hanan et al., 2015, 2014; Staver and Hansen, 2015). However, any such biases are much smaller than the bimodal signal (Staver and Hansen, 2015) and independent analyses of canopy height have confirmed that result (Xu et al., 2016, 2018).

44. This is relative and perhaps only valid up to a certain point. See effects of recent (2005 and 2014) droughts in the Amazon

That is correct. We added “up to a certain point” (line 48) to avoid confusion.

79. Most savannas in Northern South America (Venezuela and Colombia) receive more than 2000 mm year⁻¹ and they, naturally, are not forests. Perhaps this assumption needs to be validated.

It is correct that there exist very wet savannas. However, the statement in line 79 (now line 82) is not an assumption, but a result of the analysis we did on the forest cover data. This means that, although there are data points of low tree cover at rainfall levels above 2000 mm yr⁻¹, these did not lead to statistically significant bimodality in forest cover under those rainfall levels.

84. This is a critical assumption: the only potential disturbance to forest stability is rain? “Forests always recover from disturbance” Is this true, particularly in the light of current changes in which multiple disturbances occur simultaneously?

It is indeed an important assumption in our study: at mean annual rainfall levels above the point from which currently no significant amounts of savanna are found, forests—under natural disturbances—eventually have the capacity to recover. We stressed that this is an assumption by changing the text to “*we assume that forests always recover from natural disturbances*” (line 87).

98. 83 million km²? 87 million km²?

No, that would be a too large number, given that South America as a whole has an area of 17.8 million km². The estimated area of Amazon forest depends on some assumptions (such as whether the Guiana Shield is included, as in our study), but in our estimate it amounts to roughly 8.1 million km².

147. Is this really a conservative approach? I would argue that the potential to cross a tipping point (and therefore, overestimating bi-stability) is not conservative given the uncertain effects of climate change

We understand the confusion. We are conservative in our estimate of hysteresis, in the sense that if the potential analysis indicates that certain levels of mean annual rainfall *prohibit* hysteresis—even if there is a dry season with an intensity at which sometimes bistability may be found (but at other mean annual rainfall levels)—we assume no bistability at those mean annual rainfall levels. We made this conservativeness in estimating *hysteresis* more clear by changing the text to “... *a conservative approach in estimating hysteresis ...*” (lines 149-150).

151. As mentioned before, Forest-savanna transition in Northern South America does not follow the same set of rules.

We added the word “generally” so that “*those effects generally occur within the mean annual rainfall levels that define the broad-scale hysteresis of tropical forests*” (lines 153-154).

213. Fully forested forest?

Thank you for spotting this. We changed it to “fully forested continent”.

348. Any comment on the performance of ERA-Interim in the tropics (particularly in mountain regions)?

For this study we used ERA5 data, which has many improvements over ERA-Interim. Especially for the tropics, precipitation data are much improved in ERA5 (Nogueira, 2020). Furthermore, regarding large-scale dynamics and wind fields, the orography is simulated much better at 0.25° resolution (ERA5) than previously at 0.75° resolution (ERA-Interim), which has large consequences for mountainous regions as more specific spatial characteristics in these regions can be accounted for

(Belmonte Rivas and Stoffelen, 2019; Hoffmann et al., 2019). We added the following, including reference to mentioned papers: “*ERA5 has better performance than ERA-Interim regarding wind fields and rainfall, especially in the tropics*” (lines 356-357).

References

- Belmonte Rivas, M., Stoffelen, A., 2019. Characterizing ERA-Interim and ERA5 surface wind biases using ASCAT. *Ocean Science* 15, 831–852. <https://doi.org/10.5194/os-15-831-2019>
- Bonan, G.B., 2008. Forests and climate change: forcings, feedbacks, and the climate benefits of forests. *Science* 320, 1444–1449. <https://doi.org/10.1126/science.1155121>
- Bowman, D.M.J.S., Perry, G.L.W., Marston, J.B., 2015. Feedbacks and landscape-level vegetation dynamics. *Trends in Ecology & Evolution* 30, 255–260. <https://doi.org/10.1016/j.tree.2015.03.005>
- Flores, B.M., Staal, A., Jakovac, C.C., Hirota, M., Holmgren, M., Oliveira, R.S., 2020. Soil erosion as a resilience drain in disturbed tropical forests. *Plant and Soil* 450, 11–25. <https://doi.org/10.1007/s11104-019-04097-8>
- Hanan, N.P., Tredennick, A.T., Prihodko, L., Bucini, G., Dohn, J., 2015. Analysis of stable states in global savannas—a response to Staver and Hansen. *Global Ecology and Biogeography* 24, 988–989.
- Hanan, N.P., Tredennick, A.T., Prihodko, L., Bucini, G., Dohn, J., 2014. Analysis of stable states in global savannas: is the CART pulling the horse? *Global Ecology and Biogeography* 23, 259–263. <https://doi.org/10.1111/geb.12122>
- Hirota, M., Holmgren, M., van Nes, E.H., Scheffer, M., 2011. Global resilience of tropical forest and savanna to critical transitions. *Science* 334, 232–235. <https://doi.org/10.1126/science.1210657>
- Hoffmann, L., Günther, G., Li, D., Stein, O., Wu, X., Griessbach, S., Heng, Y., Konopka, P., Müller, R., Vogel, B., Wright, J.S., 2019. From ERA-Interim to ERA5: the considerable impact of ECMWF’s next-generation reanalysis on Lagrangian transport simulations. *Atmospheric Chemistry and Physics* 19, 3097–3124. <https://doi.org/10.5194/acp-19-3097-2019>
- Nogueira, M., 2020. Inter-comparison of ERA-5, ERA-interim and GPCP rainfall over the last 40 years: Process-based analysis of systematic and random differences. *Journal of Hydrology* 583, 124632. <https://doi.org/10.1016/j.jhydrol.2020.124632>
- Spracklen, D.V., Arnold, S.R., Taylor, C.M., 2012. Observations of increased tropical rainfall preceded by air passage over forests. *Nature* 489, 282–285. <https://doi.org/10.1038/nature11390>
- Staal, A., Tuinenburg, O.A., Bosmans, J.H.C., Holmgren, M., van Nes, E.H., Scheffer, M., Zemp, D.C., Dekker, S.C., 2018. Forest-rainfall cascades buffer against drought across the Amazon. *Nature Climate Change* 8, 539–543. <https://doi.org/10.1038/s41558-018-0177-y>
- Staver, A.C., Archibald, S., Levin, S.A., 2011. The global extent and determinants of savanna and forest as alternative biome states. *Science* 334, 230–232. <https://doi.org/10.1126/science.1210465>
- Staver, A.C., Hansen, M.C., 2015. Analysis of stable states in global savannas: is the CART pulling the horse?—a comment. *Global Ecology and Biogeography* 24, 985–987. <https://doi.org/10.1111/geb.12285>
- Van der Ent, R.J., Wang-Erlandsson, L., Keys, P.W., Savenije, H.H.G., 2014. Contrasting roles of interception and transpiration in the hydrological cycle-Part 2: Moisture recycling. *Earth System Dynamics* 5, 471–489. <https://doi.org/10.5194/esd-5-471-2014>
- Van Nes, E.H., Staal, A., Hantson, S., Holmgren, M., Pueyo, S., Bernardi, R.E., Flores, B.M., Xu, C., Scheffer, M., 2018. Fire forbids fifty-fifty forest. *PLoS ONE* 18, e0191027.
- Xu, C., Hantson, S., Holmgren, M., van Nes, E.H., Staal, A., Scheffer, M., 2016. Remotely sensed canopy height reveals three pantropical ecosystem states. *Ecology* 97, 2518–2521.
- Xu, C., Staal, A., Hantson, S., Holmgren, M., van Nes, E.H., Scheffer, M., 2018. Remotely sensed canopy height reveals three pantropical ecosystem states: reply. *Ecology* 99, 235–237. <https://doi.org/10.1002/ecy.2077>